# Microbial abundance, activity and population genomic profiling with mOTUs2

Alessio Milanese [1], Daniel R Mende [2], Lucas Paoli[3,4], Guillem Salazar[3], Hans-Joachim Ruscheweyh[3], Miguelangel Cuenca [3], Pascal Hingamp[5], Renato Alves [1,6], Paul I Costea [1], Luis Pedro Coelho [1], Thomas S.B. Schmidt [1], Alexandre Almeida [7,8], Alex L Mitchell[7], Robert D. Finn [7], Jaime Huerta-Cepas[1,9], Peer Bork [1,10,11,12], Georg Zeller [1] & Shinichi Sunagawa [3]

Metagenomic sequencing has greatly improved our ability to profile the composition of environmental and host-associated microbial communities. However, the dependency of most methods on reference genomes, which are currently unavailable for a substantial fraction of microbial species, introduces estimation biases. We present an updated and functionally extended tool based on universal (i.e., reference-independent), phylogenetic marker gene (MG)-based operational taxonomic units (mOTUs) enabling the profiling of >7700 microbial species. As more than 30% of them could not previously be quantified at this taxonomic resolution, relative abundance estimates based on mOTUs are more accurate compared to other methods. As a new feature, we show that mOTUs, which are based on essential housekeeping genes, are demonstrably well-suited for quantification of basal transcriptional activity of community members. Furthermore, single nucleotide variation profiles estimated using mOTUs reflect those from whole genomes, which allows for comparing microbial strain populations (e.g., across different human body sites).

[1] European Molecular Biology Laboratory, Meyerhofstr. 1, 69117 Heidelberg, Germany. [2] Daniel K. Inouye Center for Microbial Oceanography Research and Education, University of Hawai'i at Mānoa, 1950 East West Road, Honolulu, USA 96822, United States. [3] Department of Biology, Institute of Microbiology and Swiss Institute of Bioinformatics, ETH Zürich, Vladimir-Prelog-Weg 4, 8093 Zürich, Switzerland. [4] Department of Biology, École normale supérieure, 46 rue d'Ulm, 75005 Paris, France. [5] Aix Marseille Univ, Université de Toulon, CNRS, IRD, MIO UM 110, 13288 Marseille, France. [6] Candidate for Joint PhD degree from EMBL and Heidelberg University, Faculty of Biosciences, Heidelberg, Germany. [7] European Molecular Biology Laboratory, European Bioinformatics Institute (EMBL-EBI), Wellcome Genome Campus, Hinxton CB10 1 SD, UK. [8] Wellcome Trust Sanger Institute, Wellcome Genome Campus, Hinxton CB10 1SA, UK. [9] Centro de Biotecnología y Genómica de Plantas, Universidad Politécnica de Madrid (UPM) - Instituto Nacional de Investigación y Tecnología Agraria y Alimentaria (INIA), Campus de Montegancedo-UPM, 28223 Pozuelo de Alarcón, Madrid, Spain. [10] Max Delbrück Centre for Molecular Medicine, Robert-Rössle-Str. 10, 13092 Berlin, Germany. [11] Molecular Medicine Partnership Unit, Heidelberg, Germany. [12] Department of Bioinformatics, Biocenter, University of Würzburg, Am Hubland, 97074 Würzburg, Germany. These authors contributed equally: Alessio Milanese, Daniel R Mende. Correspondence and requests for materials should be addressed to G.Z. (email: zeller@embl.de) or to S.S. (email: ssunagawa@ethz.ch)

Microorganisms live in complex communities of interacting species that impact life on earth and geochemical processes in the environment. It is thus of fundamental interest to accurately profile and compare the composition of the communities they form. The most common approach for microbial community profiling is by classification of PCR amplicon sequences from the small subunit ribosomal RNA gene (i.e., the 16S rRNA gene of bacteria and archaea). While powerful, this approach is also known to introduce biases in composition estimates due to, for instance, variations in 16S rRNA gene copy numbers per genome (Supplementary Figure 1), unequal efficiencies of PCR-primers in different species[1, 2] as well as the use of different sub-regions of this gene[3]. In addition, the high level of its sequence conservation limits the power for resolving closely related organisms[4].

More recent methods sample environmental DNA directly by shotgun sequencing (metagenomics), which resolves some of these biases. Different strategies have been introduced to determine microbial community compositions from metagenomic data. One approach is based on classifying sequencing reads using publicly available and taxonomically annotated reference genome sequences of 'known' species. The resulting read abundance distributions require subsequent normalization by genome length[5, 6] to derive relative abundances of individual species (Supplementary Figure 1). Rather than using whole genomes, an alternative approach is to quantify read coverage of genes that are found to be clade-specific based on analyzing current reference genome databases[7]. If such marker genes occur only once per genome, then the resulting read coverages do not need to be normalized by copy number or genome length. However, a downside to any method depending on prior knowledge of genome sequences is that genomically uncharacterized taxa remain unaccounted for, which can lead to inaccurate relative abundance estimates at species-level resolution (Supplementary Figure 1).

Taxa that are missed by such reference-dependent methods can collectively be referred to as biological 'dark matter'[8]. These include organisms—hereon referred to as 'unknown' species—that may be detected, but remain difficult to quantify using standard methods and up-to-date genome databases. To overcome this issue, we previously introduced a profiling tool that uses universally occurring, protein coding, single copy phylogenetic marker gene (MG)-based operational taxonomic units (mOTUs) as an approach to capture and quantify microbial taxa at species-level resolution in metagenomic samples[9]. mOTUs are built on the basis of MGs from both known and unknown species, the latter of which are extracted from existing metagenomes, enabling higher taxonomic resolution and more accurate quantification of species when profiling new microbial communities[9].

Here, we present an updated and functionally extended profiling tool, the mOTU profiler version 2 (mOTUs2), which consolidates data from >3100 metagenomic samples into an updated mOTU database to substantially improve the representation of human-associated and ocean microbial species. Evaluations of mOTUs2 relative to state-of-the-art methods demonstrate improved sensitivity and quantification accuracy for both known and unknown species. We illustrate how species missed by other approaches can skew relative abundance estimates from compositional metagenomic data. Moreover, mOTUs enable quantifying baseline transcriptional activity of microbial community members from metatranscriptomic data, while avoiding quantification artefacts due to the use of non-housekeeping genes. Finally, heterogeneous populations of microbial strains have been reported in metagenomic studies to co-exist in a given microbial community, differ between individuals and environmental sampling sites, and remain stable over time[10–12]. We show that differences between such strain populations can be estimated using the MGs of mOTUs as an efficient alternative to using whole genome sequences for metagenomic single-nucleotide-variation profiling.

## Results

**Reference-extended microbial community profiling with mOTUs2.** We first identified 40 previously selected and benchmarked MGs in a total set of >25,000 sequenced genomes[13]. To obtain species-level taxonomic groups of (possibly redundant) sequences, we clustered these genomes based on a calibrated cutoff of 96.5% sequence identity[4] into 5232 non-redundant, reference MG-based operational taxonomic units (ref-mOTUs) that contained more than half of a subset of ten MGs that were found suitable for metagenomic analyses[9]. Next, we assembled >3100 metagenomes from studies that included, as a requirement, a large number of systematically processed samples per biome (Supplementary Data 1). These comprised 1210 samples from major human body sites (oral, skin, gut and vaginal[14, 15]), an additional 1693 samples from various human gut metagenomic studies including different disease cohorts[16–21] and 243 ocean water samples[22]. MGs predicted in these assemblies were clustered into marker gene clusters (MGCs). Finally, we devised an improved method for co-abundance-based binning of the MGCs into metagenomic mOTUs (meta-mOTUs) applying the same inclusion criterion (>5 MGs per mOTU) as for ref-mOTUs (Fig. 1a, Methods). To evaluate the binning accuracy of meta-mOTUs, we assessed individual MGCs in terms of taxonomic consistencies (Methods), variations in abundance, prevalence and GC-content of individual MGCs in comparison to ref-mOTUs (Supplementary Figure 2, Methods). Overall, we found high agreement in all categories. For example, at the species level, >97% (s.d.: ±1.5%) of meta-mOTUs are expected to be completely consistent in their taxonomic annotation (Supplementary Figure 2a), despite known incongruencies between species name assignments and MG-based sequence divergence[4].

After quality control, the resulting 2494 meta-mOTUs, together with the 5232 ref-mOTUs, comprise the updated mOTU database. Compared to the previous version, these numbers correspond to a 3-fold and 7-fold increase in known and unknown species, respectively, that can now be profiled using mOTUs2. Taxonomic ranks for each mOTU were assigned by a last common ancestor-based consensus assignment (Supplementary Figure 3, Methods). Also, phylogenetic reconstruction shows that meta-mOTUs were sampled from a broad taxonomic distribution (Supplementary Figure 4), including from taxa that were hypothesized to represent novel phyla[23]. Across all included biomes (four major human body sites and the ocean), the number and fraction of unknown species (85%) were highest in ocean water samples (Fig. 1b), which is in congruence with previous results[22]. Notably, even in presumably well-explored human gut samples, we found that more than half of the species still lacked sequenced representatives in our reference genome database[13] (Fig. 1b, c). A breakdown of mOTUs by biome showed that ref-mOTUs are often detected in multiple biomes, while meta-mOTUs tend to be more biome-specific (Supplementary Figures 5a, b). As shown by rank-abundance analyses, we find meta-mOTUs to be well distributed across the range from dominant to rare species (Supplementary Figure 6). Finally, the MGCs that could not be binned were used to quantify the cumulative abundance of organisms that are known to be present, but not quantified as mOTUs (Methods). This fraction was higher for the ocean than for samples from human body sites (Fig. 1c), which may be improved by increasing the number of profiled ocean metagenomes in the future.

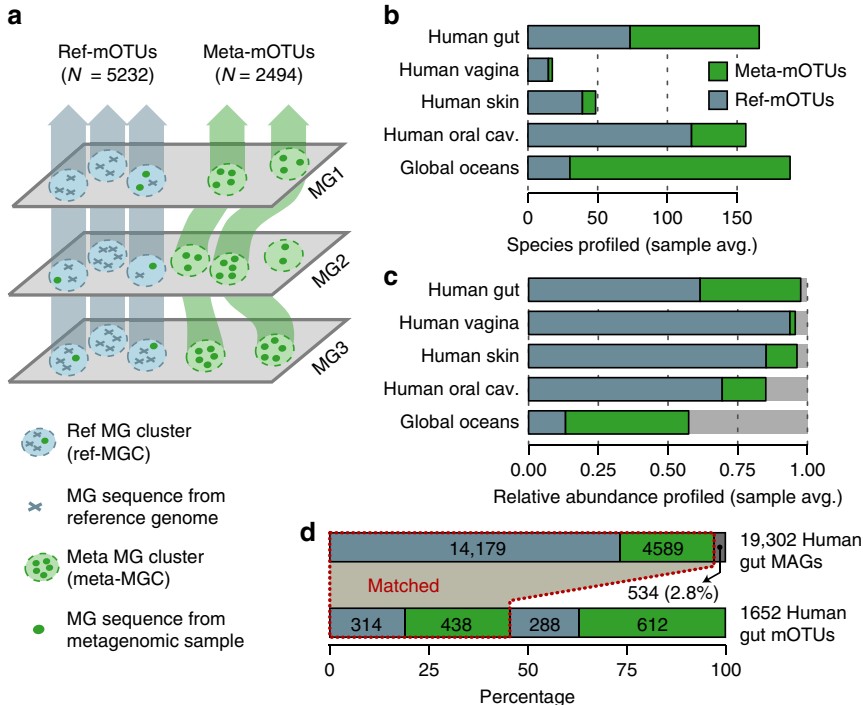

**Fig. 1** Construction of marker gene-based OTUs (mOTUs) for metagenomic profiling. **a** Schematic illustration of the mOTUs concept (Methods). **b** The observed richness of ref-mOTUs (containing exclusively MG sequences from reference genomes; blue) and meta-mOTUs (containing only MG sequences from metagenomes; green) per biome, and **c** mean cumulative relative abundance of species profiled across 2481 metagenomic samples. **d** Correspondence between mOTUs and 19,302 metagenome assembled genomes (MAGs) from the human gut. While less than 3% of MAGs are not represented (dark grey bar), mOTUs allow for profiling of 900 species not captured by MAGs. Source data are provided as a Source Data file.

We next evaluated the sensitivity of mOTUs2 for unknown species and assessed the resulting impact on relative abundance estimations compared to other approaches. To accomplish this, we analyzed the correspondence between mOTUs and metagenome assembled genomes (MAGs). MAGs involve binning assembled metagenomic contigs by sequence composition and/ or read abundance variation as a strategy to detect and genomically characterize organisms found in environmental samples[24]. Thus, similar to meta-mOTUs, MAGs may include taxa that are not yet represented in genomic databases, and thus provide a way to test if and how many environmental microbes would be captured by mOTUs. More specifically, we reconstructed MAGs from 4880 published human gut metagenomes (Supplementary Data 2) and used 1845 MAGs identified in ocean water samples as a subset of 8000 recently published MAGs[23]. Using these MAGs, we determined how many of them could be assigned to previously known (ref-mOTUs) vs. unknown species (meta-mOTUs) and evaluated the impact on relative abundance estimations. We found that >97% of MAGs from human gut samples were represented by mOTUs (Fig. 1d). Among these, 76% could be matched to ref-mOTUs and the remainder to meta-mOTUs. In addition, although the majority of the MAGs could be assigned to mOTUs, they represented only 42% of all human gut meta-mOTUs. For ocean water MAGs, 55% were represented by mOTUs (19% of these matching ref-mOTUs), while MAGs represented only 25% of ocean meta-mOTUs (Supplementary Figure 7). Our results indicate that the most abundant organisms in the human gut are already represented in public genome databases, whereas a substantial additional fraction becomes accessible through metagenomic data analysis. While assembly opens possibilities for many additional analyses, higher sequence coverage is required for the reconstruction of high-quality MAGs than for mOTUs, explaining why meta-mOTUs capture many

more species. In the ocean, even some of the most abundant species still appear to lack representative genomic information (Supplementary Figure 7).

Next, we assessed the advantage of using a reference-independent method for species quantification in microbial communities. To this end, we compared mOTUs2 with two popular reference-dependent approaches, as well as its original version (mOTUs1[9]), using: (i) simulated metagenomes from human gut-associated MAGs (Supplementary Figures 8, 9 and Methods), (ii) the Critical Assessment of Metagenome Interpretation (CAMI) dataset[25] (Supplementary Figures 10, 11), and (iii) the simulated metagenomes used to evaluate MetaPhlan2[7] for benchmarking (Fig. 2; Supplementary Data 3, 4; Supplementary Table 1). Our results based on simulated MAGs show that in terms of precision, mOTUs2 and MetaPhlan2 outperformed mOTUs1 and Kraken (Fig. 2e). The fact that the reference-dependent methods MetaPhlan2 and Kraken can only detect genomes that are closely related to those present in current reference databases was well reflected in a reduced sensitivity, higher mean absolute error and deviations from expected taxonomic diversity estimates (Fig. 2e–g). Additional simulations showed that relative abundance estimates may be highly inaccurate when solely relying on reference genomes if unknown species are present in medium to high abundance (Supplementary Figures 9, 11). For the CAMI dataset, our results show that the mOTUs2 profiler outperformed many other tools (Fig. 2h–o; Supplementary Figures 10, 11). More specifically, mOTUs2 not only outperformed mOTUs1 at all taxonomic ranks, but also other tools, including MetaPhlan2 above the genus level for medium complexity simulations and above the species level for high complexity samples (Fig. 2k, o). We should note that in the CAMI benchmark (and the OPAL evaluation tool[26]) profiled abundance data are re-normalized based on the detected taxa (see

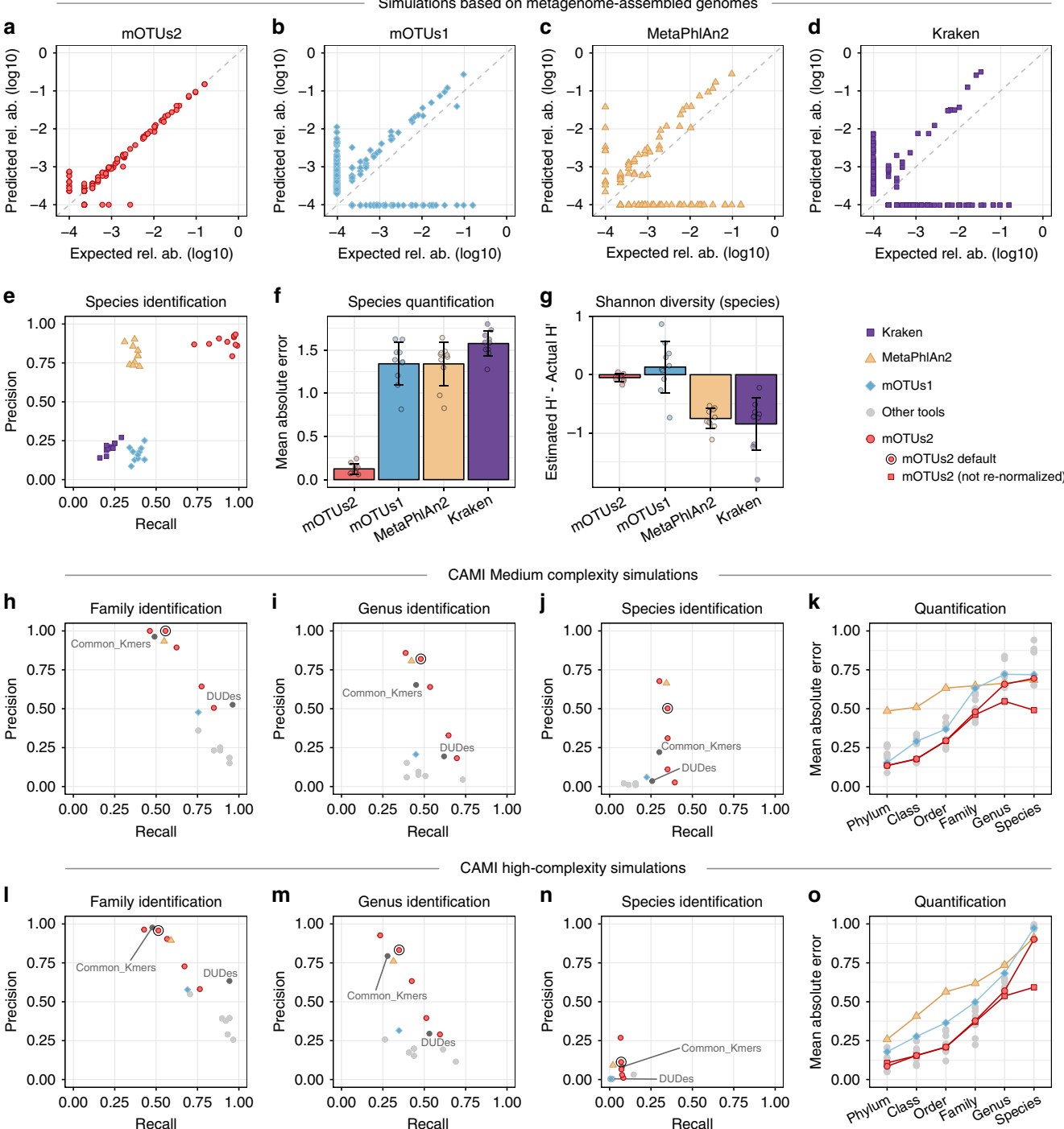

**Fig. 2** Evaluation of mOTU profiling on simulated samples. Benchmarks of quantification accuracy (**a**–**g**) on ten simulated metagenomic samples (Methods) containing MAGs with ($n = 50$) and MAGs without ($n = 50$) a representative reference genome sequence, (**h**–**o**) and the CAMI challenge datasets[25]. **a**–**d** A representative simulated metagenome (out of ten; Supplementary Figures 8, 9) analysed with four profilers. **e** Precision-recall plot, where each data point corresponds to one of the ten simulated samples. Mean absolute error (MAE, also referred to as L1 norm) (**f**) and differences of the Shannon diversity index (**g**) from the expected values (error bars in f and g show standard deviation). **h**–**j** Average precision-recall values over the two medium complexity samples and (**l**–**n**) average precision-recall values over the five high complexity samples of the CAMI dataset (see also Supplementary Figure 10). Each precision-recall plot contains five values for mOTUs2, which correspond to different sets of parameters: high precision (-l 140 -g 6), default (-l 100 -g 3), recall (-l 75 -g 3), high recall (-l 50 -g 2) and maximum recall (-l 30 -g 1), indicating the versatility of mOTUs2 in optimising precision or recall. In (**k**) and (**o**), mean absolute errors (MAE; referred to as L1-norm in CAMI) at different taxonomic ranks are shown for several tools. For mOTUs2, results for two options of calculating relative abundances are shown: one with relative abundances re-normalized based on detected taxa, which is enforced in the CAMI evaluation (but artificially deteriorates quantification accuracy), and one without this additional re-normalization (see main text and Supplementary Figure 11 for details). Data are provided in Supplementary Data 3, 4. Other taxonomic profilers (MetaPhyler, TIPP, Taxy-Pro, FOCUS, CLARK, Quickr) evaluated in CAMI[25] are denoted by grey dots. Source data are provided as a Source Data file.

Supplementary Figure 11b). This re-normalisation procedure penalises tools, such as mOTUs2, that can account for the relative abundance of unknown taxa (Supplementary Figures 1, 11b). This feature leads to improved quantification (hence, a further reduction of the mean absolute error), in particular at the species level (Fig. 2k, o; Supplementary Figure 11a). Finally, since Kraken was not included in the CAMI benchmark[25] dataset, we compared the performance of mOTUs2 to the results reported for the evaluation of MetaPhlan2[7], which included Kraken[6]. We find that mOTUs2 and MetaPhlan2 performed similarly, while both (and mOTUs1) outcompeted Kraken (Supplementary Table 1).

Given that profiling unknown species in addition to those represented in genome databases significantly improves relative abundance estimates, we sought to assess potential impacts on describing community structural properties. The total number of detected species and their relative abundance distribution determines the alpha diversity of a microbial community. This parameter is of fundamental interest in microbial ecology including in studies of gastrointestinal diseases[27]. As the quantitative breakdown of unknown species into mOTUs provides more accurate estimates of relative species abundances, measures for alpha diversity, such as the Shannon index ($H'$), were expected to be more accurate for mOTU-based profiles compared to reference-dependent approaches (based on simulations, Fig. 2g). To test this further using real microbial community data, we compared mOTUs2 to reference-dependent methods against 16S rRNA gene-based approaches. In two example data sets, one from a colorectal cancer study[21] ($n = 129$) and one from an ocean ecosystem survey[22] ($n = 139$), we found mOTUs2 profiles to have higher correlations with 16S rRNA gene-based estimates of alpha diversity (Spearman $R = 0.71$, $P < 0.0001$ and $R = 0.78$, $P < 0.0001$, respectively) than the reference-dependent methods (Fig. 3 and Supplementary Figure 12).

We also assessed the performance of methods at estimating how similar taxonomic compositions are between samples (beta diversity). For this, we used data from healthy individuals who donated samples from four different major body sites on multiple sampling occasions[14], so that composition similarities could be compared within and between individuals. Given that compositional differences are expected to be smaller within than between individuals[14], we tested in how many cases a sample from one subject would be most similar to another sample from the same individual (and body site) than from any other sample in the set of >1200 samples tested. As a result, we found that mOTUs2 performed similarly to the reference-dependent, clade-specific gene-based method[7], while both outperformed the whole genome-based method used by Kraken[6] (Supplementary Figure 13).

**Unbiased metatranscriptomic profiling using marker genes.** Although metagenomics data can be used for taxonomic profiling of microbial communities, it does not allow determining whether community members are physiologically active or not. Analogous to DNA for metagenomics, metatranscriptomics refers to the sequencing of reverse-transcribed RNA present in a microbial community. Depending on environmental conditions, the number of transcripts per cell varies for most genes. An exception to this are housekeeping genes that are expressed constitutively and with low variability under different conditions. Thus, the abundance of transcripts from such genes should strongly correlate with the abundance of active cells in a community. As all ten MGs are universal and involved in the highly conserved process of translating mRNA to proteins, we hypothesized that metatranscriptomic abundances would serve as particularly good proxies for relative cell abundances. To test this, we compared mOTUs2 to reference-dependent methods that have been used in recent metatranscriptomic studies[28, 29] or analysis workflows[30] relating metatranscriptomic profiles to microbial abundance and/or activity. More specifically, we correlated matching metagenome and metatranscriptome profiles from human stool samples[31]. At the species level (Fig. 4a, Supplementary Figure 14), mOTUs2-based correlations were considerably higher (median Spearman's $R = 0.76$) than for reference-dependent methods ($R = 0.37$ and $0.45$). Furthermore, we summarized mOTU abundances at the class level and computed all pairwise distances for all metagenomic and metatranscriptomic profiles to test for each metagenomic profile whether the most similar metatranscriptomic profile matched the same sample. For mOTUs2, this was the case for 92% of the samples compared to 78% and 64% for reference-dependent methods (Fig. 4b, Supplementary Figure 15).

**MG-based SNV profiling for microbial population analyses.** Originally, the ten MGs were identified as a subset of candidate phylogenetic marker genes deemed suitable for reconstructing the tree of life[32] due to their universal occurrence and low rate of horizontal gene transfer[33]. These properties provided us with the opportunity to test how well single nucleotide variation (SNV) profiles of microbial populations could be recapitulated by the MGs comprising mOTUs as a compute-efficient alternative to using whole reference genome sequences. To this end, we generated metagenomic SNV profiles[34] for sets of samples from different human body sites and ocean water using ref-mOTUs and representative genome sequences as reference databases. Despite some differences between biomes (Fig. 5a) and a few species, we found overall that the distances of SNV profiles using MGs were highly correlated ($R > 0.8$; Pearson) with those obtained using whole genomes. For example, we find almost perfect correlations for ocean microbial species (median $R = 0.96$), and for most gut microbial species (median $R = 0.84$) including those for which sub-species population structure was recently identified[12, 15, 35] (Supplementary Figure 16).

Having established the possibility of resolving mOTUs below the species level, we addressed the question of how variable

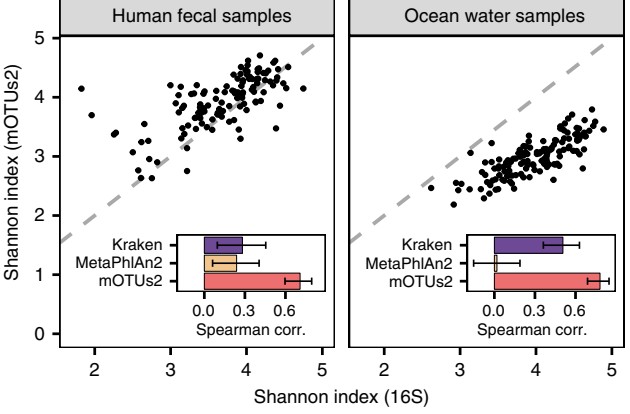

**Fig. 3** Reference-extended mOTUs for microbial community diversity profiling. Shannon index was calculated based on 16S rRNA gene (16S) fragments (x-axis) and mOTUs (y-axis), respectively, for 129 human faecal samples (left) and 139 ocean water samples (right). Mean Spearman correlation of diversity estimates based on 16S and three metagenomic profiling tools (Kraken, MetaPhlAn2 and mOTUs2) are shown in the insets. Error bars delineate 95% confidence intervals after bootstrapping. Source data are provided as a Source Data file.

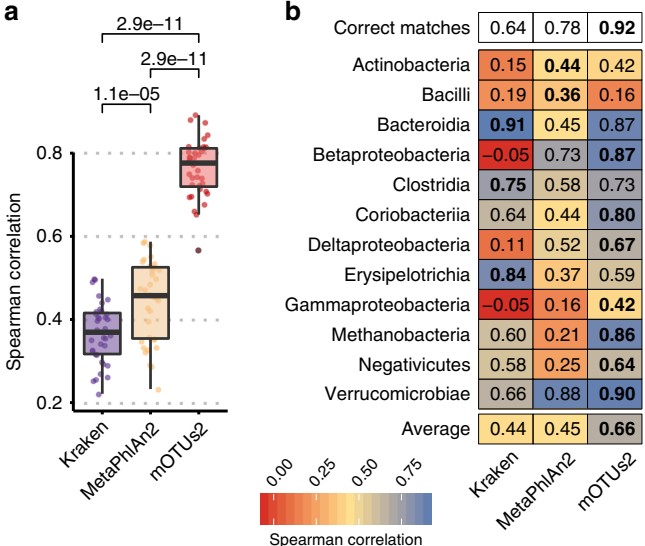

**a**

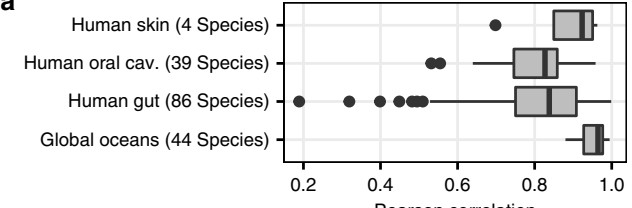

**b**

| | Kraken | MetaPhlAn2 | mOTUs2 |
|---|---|---|---|
| Correct matches | 0.64 | 0.78 | **0.92** |
| Actinobacteria | 0.15 | **0.44** | 0.42 |
| Bacilli | 0.19 | **0.36** | 0.16 |
| Bacteroidia | **0.91** | 0.45 | 0.87 |
| Betaproteobacteria | −0.05 | 0.73 | **0.87** |
| Clostridia | **0.75** | 0.58 | 0.73 |
| Coriobacteriia | 0.64 | 0.44 | **0.80** |
| Deltaproteobacteria | 0.11 | 0.52 | **0.67** |
| Erysipelotrichia | **0.84** | 0.37 | 0.59 |
| Gammaproteobacteria | −0.05 | 0.16 | **0.42** |
| Methanobacteria | 0.60 | 0.21 | **0.86** |
| Negativicutes | 0.58 | 0.25 | **0.64** |
| Verrucomicrobiae | 0.66 | 0.88 | **0.90** |
| Average | 0.44 | 0.45 | **0.66** |

**Fig. 4** Metatranscriptomic abundance profiling with mOTUs2. **a** Spearman correlation between matched metagenomic and metatranscriptomic profiles obtained from 36 faecal samples with Kraken, MetaPhlAn2 and mOTUs2. mOTUs2 profiles (red) show significantly higher correlation than the other two methods (paired two-sided Wilcoxon test, boxplots show the median correlation as horizontal lines and interquartile ranges as boxes, whiskers extend at most 1.5 times the interquartile range). **b** The top-row represents the proportion of cases in which the distance (log-Euclidean) between metagenomic and metatranscriptomic profiles was smallest for the same sample. Below is a taxonomic breakdown (12 most abundant classes) of correlations between metagenomic and metatranscriptomic profiles. For each class, the highest correlation value across the tested methods are shown in bold. Source data are provided as a Source Data file.

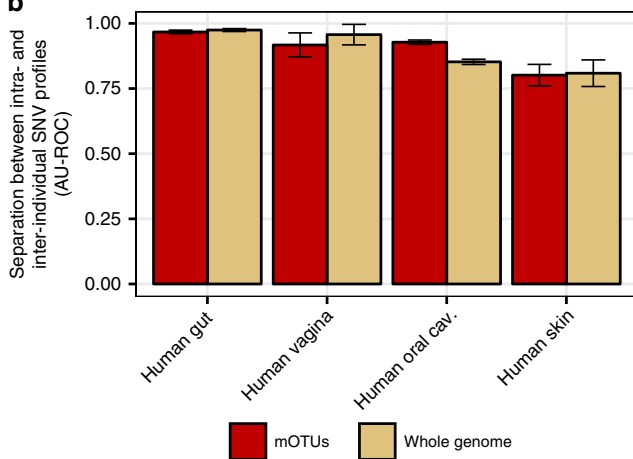

**Fig. 5** Marker gene-based SNV profiles are comparable to those using whole genomes. **a** Pearson correlation coefficients for MG- and genome-based SNV profiles across species and biomes in the HMP ($N = 2807$) and ocean dataset ($N = 139$). Median correlations (Pearson's r) are shown as horizontal lines and interquartile ranges as boxes. Whiskers extend at most 1.5 times the interquartile range. **b** Intra- and inter-individual distances of SNV profiles were compared using the area under the receiver operating characteristic curve (AU-ROC) to determine the degree of individuality of microbial strain populations for different human body sites (see also Supplementary Figure 17). Error bars delineate 95% confidence intervals after bootstrapping. Source data are provided as a Source Data file.

microbial strains populations were over time in different human body sites. Previously, microbial strain populations were shown to display a high degree of individuality e.g., in the gut, skin, and oral sites[11, 36, 37]. However, a comparative analysis of the degree of individuality of strain populations across different human body-sites has not yet been performed. Using both ref-mOTUs and meta-mOTUs, we compared strain population similarities of body site samples collected in the HMP project[14, 15] and found that stool and vaginal samples display the highest degree of individuality, followed by oral and skin samples. Again, these results were highly consistent with those obtained for reference genomes (Fig. 5b and Supplementary Figure 17).

## Discussion

The original development of the mOTU profiler was driven by the motivation to extend reference-dependent profiling of human gut microbial species to uncharacterized taxa. As more environments are subjected to metagenomic profiling, more data sets are becoming available that can be used for approaches based on binning genes by co-abundance analysis. With the inclusion of new microbiomes, we found that some human body sites are very well represented by available reference genomes (in particular skin and vagina). In contrast, more than 50% of gut microbial species still lack representative reference genomes (see also ref. [38]), which may seem unexpected, but this estimate is in the same range as reported for an independent approach[39]. This may in part be due to methodological improvements in the binning of MGs into meta-mOTUs (Methods), increasing the number of potentially uncharacterized species that can be profiled. In addition, we included not only more samples, but also data from a

number of disease-related studies (e.g., CRC, liver cirrhosis, type 2 diabetes) with large geographic distribution both contributing to an extended diversity of species that were not profiled previously. These may include species of particular relevance for differentiating healthy from diseased states. Furthermore, our results highlight the critical need to generate more reference genomes for the ocean environment where we find only 15% of species to have a representative genome sequenced. Future efforts could aim at extracting MGs from high-quality MAGs and single amplified genome sequences to incorporate these into the mOTU database.

Although metagenomics data can be used to profile the abundance of microbial taxa in a given community, they do not inform us as to whether they are also (transcriptionally) active. To discern genomic potential from activity, the combined use of metagenomics with metatranscriptomics profiling is becoming increasingly popular. Here, we found that metatranscriptomic abundances of mOTUs are highly correlated with metagenomic abundances, which highlights the property of MGs as constitutively expressed housekeeping genes across different conditions. This suggests that mOTUs should be useful for normalizing metatranscriptomics data for differential gene expression analyses. Other methods depending on genes that are conditionally or variably expressed are demonstrably less suitable for this approach and may also give the misleading impression that many taxa are rare, but highly active, or abundant, but inactive or dead (Supplementary Figure 15).

The computation of metagenomic SNV profiles to study microbial strain population differences is both resource and time-consuming when using methods based on whole reference genome sequences[34, 38]. We show that the use of mOTUs provides a fast and efficient alternative for profiling abundant species in microbial communities. In addition to the improved efficiency, mOTUs enable studying differences in strain populations of species that currently lack a representative genome sequence. This may be particularly relevant for disease-associated species and biomes for which only few reference genomes are available. A breakdown of intra-individual strain population similarity by species also allows for distinguishing those with high specificity, potentially under the control of the immune system, from those that only transiently populate their host. Promising applications of this approach could include testing the efficacy of strain-retention after faecal microbiota transplantation[10] or studying dispersal patterns of microbial populations in the environment.

## Methods

**The mOTUs2 profiler**. The mOTU profiler version 2 (mOTUs2) is a stand-alone, open source, computational tool that estimates the relative abundance of known as well as genomically uncharacterized microbial community members at the species level using metagenomic shotgun sequencing data. The taxonomic profiling method is based on ten universally occurring, protein coding, single-copy phylogenetic marker genes (MGs), which were extracted from more than 25,000 reference genomes[13] and more than 3100 metagenomic samples (Supplementary Data 1; in total ca. 367,000 non-redundant MG sequences). The MGs were grouped into >7700 MG-based operational taxonomic units (mOTUs) that represent microbial species, many of which (ca. 30%) still lack sequenced reference genomes. In addition to (i) taxonomic profiling, the tool allows for (ii) basal transcriptional activity profiling of community members using metatranscriptomic data as well as (iii) determining proxies for strain population genomic distances based on single-nucleotide variations (SNVs) within the phylogenetic marker genes that comprise mOTUs.

**Generation and annotation of the mOTUs2 database**. The mOTUs2 profiler relies on a custom-built database of MG sequences extracted from reference genomes (ref-MGs) and from metagenomic samples (meta-MGs). The reference genomes were grouped into species-level clusters (specI clusters) and MG sequences from these reference genomes were grouped based on their specI affiliation into reference marker gene clusters (ref-MGCs). These ref-MGCs were augmented by meta-MGs and the remaining meta-MGs were clustered into meta-MGCs. MGCs of different MGs were subsequently grouped based on their specI affiliation or binned based on co-abundance analysis into reference genome-based mOTUs (ref-mOTUs) and metagenomic mOTUs (meta-mOTUs), respectively. The resulting mOTUs were quality-controlled, compiled into a sequence database for short-read mapping and taxonomically annotated. Regular updates of the of the mOTU database will be made available at: http://motu-tool.org.

**Collection of MGs from reference genomes and metagenomes**. The 25,038 reference genomes used for the mOTU database were downloaded from the pro-Genomes database[13]. Metagenomic data were downloaded from the Genbank Sequence Read Archive (https://www.ncbi.nlm.nih.gov/sra) and the European Nucleotide Archive (https://www.ebi.ac.uk/ena) (accession numbers are listed in Supplementary Data 1). Most samples were obtained from human microbiome studies, including 1210 samples from different major human body sites (oral, skin, gut and vaginal[14, 15] and 1693 further samples from various human gut microbiome studies[16–21]. In addition, we used 243 metagenomic samples from an ocean microbiome study[22]. All samples were processed for marker gene identification[9]. Briefly, quality-controlled raw sequencing reads were subjected to metagenomic assembly and genes predicted on contiguous sequences longer than 500 base pairs (bp). MGs were subsequently extracted using the fetchMGs tool (available at http://motu-tool.org/fetchMG.html). In short, fetchMGs identifies MGs using HMM models built with HMMER3 (http://hmmer.org) applying a set of optimized cutoffs[4, 9], and extracts corresponding nucleotide sequences with the Seqtk tool. With this workflow we extracted a set of 40 MGs (COG0012, COG0016, COG0018, COG0048, COG0049, COG0052, COG0080, COG0081, COG0085, COG0087, COG0088, COG0090, COG0091, COG0092, COG0093, COG0094, COG0096, COG0097, COG0098, COG0099, COG0100, COG0102, COG0103, COG0124, COG0172, COG0184, COG0185, COG0186, COG0197, COG0200, COG0201, COG0202, COG0215, COG0256, COG0495, COG0522, COG0525, COG0533, COG0541, COG0552)[32, 33] from all 25,038 reference genomes. Not all of these genes are currently suitable for metagenomic applications due to high rates of ambiguous mapping of short reads owing to highly conserved regions within MG sequences as well as lower assembly rates observed for some MGs[4, 9]. Hence, a

selected subset of ten MGs (COG0012, COG0016, COG0018, COG0172, COG0215, COG0495, COG0525, COG0533, COG0541, COG0552) was extracted from genes that were predicted in metagenomes as described above.

**Grouping of MGs into ref-MGCs and meta-MGCs**. Reference genomes were processed and clustered into specI clusters to build ref-MGCs[4]. To this end, we calculated pairwise global nucleotide identities for all genome for each of the 40 MGs using vsearch (version v1.9.3)[40]. Genome-to-genome distances were calculated as the gene length-weighted arithmetic mean of the individual MG sequence distances. The resulting distance matrix was used as input for average linkage clustering using an optimized cutoff of 96.5% nucleotide identity[4], resulting in 5306 specI clusters. To assess the quality of grouping genomes into specI clusters, we tested whether the taxonomic annotations of the individual genomes provided by the NCBI were congruent (Supplementary Figure 18). More specifically, all specI clusters were annotated taxonomically in accordance to their member genomes. SpecI clusters were either homogeneous (all members had the same species-level annotation), heterogeneous (different species annotations found in the same cluster) or undetermined (clusters only containing genomes with non-binomial species names such as: *Synechocystis* sp. PCC 6803). We further evaluated how many NCBI species names occurred multiple times (in different clusters). Subsequently, the ten MGs suited for metagenomics were extracted from the specI clusters resulting in over 51,000 ref-MGCs.

To enable the profiling of species that are not yet represented by reference genomes, we extracted MG sequences from metagenomic assemblies using the fetchMGs tool. For clustering, we first calculated all pairwise distances between MGs from ref-MGs and meta-MGs using vsearch (version v1.9.3)[40] and retained alignments of at least 20 aligned bases. Then, we used open-reference clustering (employing the average linkage hierarchical clustering algorithm) to augment the pre-existing ref-MGCs with meta-MGs. The remaining meta-MGs sequences were clustered into meta-MGCs containing only meta-MGs.

**Binning of MGCs into mOTUs**. As the clustering of meta-MGs into meta-MGCs was performed independently for each of the ten MGs, it resulted in unbinned meta-MGCs (as opposed to the ref-MGCs, which were grouped into mOTUs based on their specI cluster affiliation). In order to bin MGCs into mOTUs (i.e., to link MGCs originating from the same species), we utilized the property that genes (and therefore, MGCs) from the same species are expected to co-vary in abundance across metagenomic samples[41]. Accordingly, we calculated the correlation between pairwise MGC abundances across all samples for each biome. We optimized the correlation measure and prevalence filtering (as a means against the spurious correlation between low-prevalence MGCs, see[9]) for each biome separately based on the AU-ROC determined by cross-validating the grouping of ref-MGCs for which membership in the same specI cluster served as a ground truth. As a result, we defined the following biome-specific parameters: human gut - prevalence filter: five samples, Pearson correlation of log-transformed relative abundance; ocean - prevalence filter: five samples, Pearson correlation of relative abundance; human oral cavity - prevalence filter: 50 samples, Pearson correlation of relative abundance; human vagina - prevalence filter: five samples, Pearson correlation of log transform relative abundance; human skin - prevalence filter: ten samples, Spearman correlation of log-transformed relative abundance. In order to combine the biome-specific correlations we transformed each of these into an FDR-calibrated association measure in such a way that for a given FDR value, the same association value was assigned. To obtain a single measure of association for each pair of MGCs, we computed the maximum of the FDR-calibrated association values across biomes.

For the actual binning, we used a slightly modified version of the greedy algorithm described in ref. [9]. As an initialization step, the ref-MGCs were grouped according to their specI cluster affiliations. Then, meta-MGCs were progressively binned starting from the highest FDR-calibrated association values and decreasing until a cutoff value of 0.8 was reached. In this procedure, an MGC was added (binned) to an existing group (or another MGC to form a bin of size two) if this MG (among the ten possible ones) was not already present. Only groups with at least 6 MGCs were retained and defined as mOTUs, which resulted in 2494 meta-mOTUs (consisting only of meta-MGCs) and 5232 ref-mOTUs (containing at least one ref-MGC and possibly additional meta-MGCs). MGCs that remained unbinned were grouped into a single unbinned group. Note that although specI clusters and ref-mOTUs are conceptually similar, there are two major differences: first, ref-mOTUs are composed of MGCs of at least six out of the ten different MGs used for metagenomics, while specI clusters represent genomes that are grouped based on distances calculated from up to 40 MGs; second, ref-mOTUs can, as described above, contain MGs and MGCs that were assembled from metagenomic samples.

To assess the expected taxonomic consistency of the binning strategy of meta-MGCs, a fraction of the ref-MGCs were treated in the same ways as meta-MGCs and their taxonomic affiliation (known from ref-mOTU membership) was only used afterwards to ascertain the error rate of the binning algorithm (Supplementary Figure 2a). Across all metagenomic samples used to construct the mOTUs, 1223 ref-mOTUs were detected and could be used for 100-fold resampled 5-fold cross-validation. We also assessed the agreement of the MGCs for each mOTU in terms of relative abundance and prevalence across metagenomic samples (Supplementary

Figures 2b,c). Relative abundance and prevalence showed higher agreement for meta-mOTUs than for ref-mOTUs. This was expected since the binning algorithm is directly influenced by these two parameters. We additionally evaluated the homogeneity of GC content among the MG sequences within each mOTU (Supplementary Figure 2d). meta-MGCs showed very homogeneous GC content, as expected for genes that originate from the same genome, but not for erroneously binned MG sequences.

**Construction of the mOTUs2 mapping database**. We compiled a sequence database against which short metagenomic reads can be aligned to quantify the abundance of MGCs and mOTUs. To construct a non-redundant mOTUs mapping database, we removed identical MG sequences. MG sequences in the database were extended at the start and end of the gene by up to 100 nt, based on their genome or metagenomic assembly of origin, to reduce known mapping artifacts at gene boundaries. The resulting non-redundant database consists of the sequence files in FASTA format along with MGC and mOTU annotations, as well as the coordinates of the coding segments of the MG sequences. The sequence files were further indexed for searches with BWA[42]. For SNV calling, we constructed an additional database that only consists of the centroid (medoid) sequence of every MGC so that SNVs can be identified with respect to one reference sequence per MGC.

**Taxonomic annotation of meta-mOTUs**. To assign taxonomic affiliations to meta-mOTUs, we first annotated each MG using Uniprot's UniRef90 (https://www.uniprot.org/uniref, release 2017_08) as a reference protein sequence database[43], which was supplemented with a set of additional marine protein sequences as described in[44]. Similarities between translated MG sequences and reference database entries were computed using MMSEQS2[45] with the following parameters: search -a true -e 1E-5 --max-seqs 1000. Taxonomic affiliation was assigned using a weighted Lowest Common Ancestor (LCA) approach as follows: for each MG, all protein matches in the reference database with a value ≥90% of the highest bitscore were kept. Then, outlier taxa were excluded by using a bitscore-weighted LCA annotation that covered at least 75% of the sum of all bitscores of each MG. Next, we transferred the annotation of the best-scoring MG member to each MGC and used the MGC annotations to assign a taxonomy to meta-mOTUs as follows: for each meta-mOTU and for each taxonomy rank, we required at least three MGCs to be annotated to consider the meta-mOTUs as annotated at this rank. Annotated meta-mOTUs were considered consistent if at least half of the MGC taxonomy annotations were in agreement.

**Phylogenetic analysis of mOTUs**. To explore the phylogeny of mOTUs (ref-mOTUs and meta-mOTUs), a reference tree was reconstructed by combining the phylogenetic signal of the ten sets of marker genes selected (Supplementary Figure 4). For this, all marker genes were translated into amino acid sequences and analyzed using ETE Toolkit v3.1. 1[46]. In particular, the program *ete-build* was used to run the following phylogenetic workflow: First, each set of marker proteins was independently aligned using ClustalOmega[47]. Next, alignment columns with less than three aligned residues were removed. Finally, the ten individual MG alignments were concatenated and used to infer a maximum likelihood phylogenetic tree using IQTree[48] and the LG model.

**The mOTUs2 profiling workflow**. The mOTUs2 workflow for taxonomic profiling consists of three steps: alignment of metagenomic sequencing reads to MGs, estimation of read abundances for every marker gene cluster (MGC), and calculation of mOTU abundances. As input, mOTUs2 expects the user to provide quality controlled sequencing reads. These are aligned to the MGs of the mOTU database using BWA (mem algorithm, default parameters)[42]. The resulting alignments are filtered and only those with at least 97% nucleotide identity are retained. Further, alignments are filtered according to their lengths (default: 75 bp minimum alignment length; can be adjusted using the -l option).

Next, we compute the best alignment(s) for every insert (read pair) to the MGCs using BWA alignment scores. Inserts with a single highest scoring alignment are flagged as "unique alignments", whereas inserts with multiple highest scoring alignments are flagged as "multiple alignments". Subsequently, abundances for each MGC are calculated by summing up the number of all inserts flagged as unique alignments resulting in a unique alignment profile. Inserts flagged as multiple alignments are distributed among their best-scoring MGCs in accord with their respective abundances estimated based on the unique alignment profile. Thus, the final abundances are calculated as the sum of the unique abundance profiles and the distributed contributions of the inserts flagged as multiple alignments. In addition to these MGC insert counts, MGC base coverages are calculated by first summing up the total number of bases aligning to each MGC and then dividing by the respective gene lengths. Finally, the abundances of the mOTUs are calculated as the median of their respective MGC abundances (insert counts and base coverages). In order to reduce false positive results, we require a certain number of MGCs to be detected, that is to have metagenomic reads mapped to them (default: 3 MGs, -g option in mOTUs2). Although mOTUs2 is able to profile many organisms not yet represented by reference genomes, there are still around 25% of the MGCs that could not be

binned into mOTUs (see section 2.5). Reads mapping to those MGCs are assigned to a group labelled as "unbinned" (shown as "-1" in mOTU abundance profiles). The abundance of this group is calculated as the median of unbinned MGCs summed by COG.

**Description of taxonomic profiling outputs**. The mOTUs2 profiler returns multiple taxonomic profiles, since abundances based on read mappings can be calculated in different ways. One major distinction is the unit of counts. Either fragments such as inserts (or reads for single-pair sequencing) or mapped base-pairs can be counted. Counting the mapped base-pairs has the advantage that the mean base coverage can easily be computed by dividing the number of bases aligned to a certain gene by its corresponding length (mOTUs2 output -y option: "base.coverage"). Count based statistics are powerful for differential abundance testing (output -y option: "insert.raw_counts"). As the counts could in principle be non-integer numbers due to inserts mapping to multiple genes (see section 3.1), all counts are rounded to integers. For relative abundance-based estimates, gene-length normalizations are required to account for varying lengths of MG sequences and varying numbers of MGCs present in each mOTU. To this end, we previously introduced "scaled counts" that retains most of the characteristics of insert counts. In this approach, coverages are calculated as described above and are then normalized to sum up to the number of inserts that align to MGCs (output -y option: "insert.scaled_counts").

**Single-nucleotide variant analysis with MGs**. The mOTUs2 profiler has new functionality to compute metagenomic SNV profiles using the MGs comprising mOTUs as reference sequences. The resulting SNV profiles are highly correlated to those obtained by whole genome SNV profiling (see main text, Fig. 5, Supplementary Figures 16, 17). The overall SNV calling pipeline starts by aligning metagenomic sequences to centroid sequences of MGCs (see above), before the resulting bam files are post-processed using metaSNV functions[34]. The mOTUs2 command map_snv maps the reads using BWA[42] and performs read filtering in a similar fashion as described for taxonomic profiling. For the SNV analyses, only inserts flagged as unique alignments are kept and the resulting sam file is sorted and converted into a bam file. Using the snv_call command, the tool (i) computes base coverages, (ii) calls SNVs, (iii) generates filtered allele frequency tables, and (iv) calculates distances between strain populations.

These four steps are directly built upon metaSNV capabilities[34], although the procedure was adapted to mOTUs2 to facilitate its use with genes rather than genomes. Firstly, each bam file is processed to compute per sample coverages for every reference sequence/mOTU, both vertical (average number of reads per position) and horizontal (percentage of the sequence covered at least once). SNVs are subsequently called using samtools mpileup[49], followed by two post-processing steps. This includes a filtering step, which was modified to include parallelized computing capabilities as well as the removal of padded regions in the allele frequency tables. The filtering parameters remain identical, with updated default values to account for the universal character of the genes considered: (-fb) minimal percentage of the sequence horizontally covered per sample and per mOTU (default = 80), (-fd) minimal average vertical coverage per sample and per mOTU (default = 5), (-fm) minimum number of samples meeting the listed criteria per mOTU (default = 2), (-fc) minimum vertical coverage per SNV position (default = 5), (-fp) minimum proportion of samples meeting the previous criterion at said position (default = 0.9). Finally, the filtered allele frequency tables are used to compute genetic distances between samples for each mOTU, Manhattan distances as well as major allele distances are used as the population genomic distance measure. For the latter, only allele frequency changes above 50% between the two samples are taken into account. The mOTU profiler uses parallelized computing capabilities for this step.

The output directory (-o) includes three files: two with the coverage information for each mOTU, both horizontal (*.cov.tab file) and vertical (*.perc.tab file), and a log file. Additionally, there are two directories: (i) per mOTU filtered allele frequencies of identified SNVs across samples (filtered-* directory) and (ii) per mOTU genetic distances between samples (distances-* directory), both Manhattan (mann.dist files) and major allele (allele.dist files).

**Benchmarking mOTUs2 against other tools**. To evaluate its accuracy and robustness, we benchmarked mOTUs2 against two established tools for taxonomic profiling of metagenomic samples: MetaPhlAn2[7], which is based on clade-specific marker genes, and Kraken[50], which is based on exact alignments of genomic k-mers. MetaPhlAn2 (version 2.6.0) was executed with default parameters. For Kraken-labelled analyses, we executed Kraken for read classification and calculated relative abundances with Bracken[6]. Kraken and Bracken were installed as version 1.0.0 using conda. The Minikraken database (version mini-kraken_20171101_8GB_dustmasked) was downloaded from https://ccb.jhu.edu/software/kraken/. The Minibracken database was downloaded from https://ccb.jhu.edu/software/bracken/ on 1 February 2018. We executed kraken using paired-end and single-end data using default parameters. Abundance estimation with Bracken was performed with the following parameters: -k minikraken_8GB_75mers_distrib.txt -l S -o result.abundance.bracken.

**Comparison of mOTUs with metagenome-assembled genomes**. We further validated the mOTUs using metagenome-assembled genomes (MAGs) reconstructed from different environments. For this purpose, we first extracted 4880 metagenomic sequencing runs from human gut samples available from the European Nucleotide Archive (accession numbers are listed in Supplementary Data 2). Raw reads from each run were assembled using metaSPAdes v3.10.0[51] and subsequently binned with MetaBAT2 (v2.12.1)[52] with a minimum contig length threshold of 2000 bp. Sequencing coverage required for binning was inferred by mapping the raw reads back to the assemblies using BWA v0.7.16[42] and then retrieving the percentage of mapped read bases with samtools v1.5[49] and the jgi_summarize_bam_contig_depths function from MetaBAT2. Quality scores (QS) of each metagenome-assembled genome (MAG) were estimated with CheckM v1.0.7[53], calculated as the level of completeness - 5 x contamination, as previously described[23]. Good-quality MAGs (QS > 50) were kept for subsequent downstream analyses. MAGs from marine samples (Ocean MAGs) were obtained as a subset of about 8000 MAGs, which are described in a recent publication[23]. In order to identify ocean-associated MAGs, we first searched for the keywords: ocean, marine, baltic sea and north sea to extract entries in Supplementary Table 1 of[23] and found 400 samples matching these keywords. From these samples, we selected 1845 MAGs (from Supplementary Data 2) that were reconstructed from these metagenomes.

Correspondence between MAGs and mOTUs was established using the following procedure: first, we extracted the ten MGs from the MAGs using fetchMGs (see above), obtaining a set of MG-MAGs. Second, we aligned the MG-MAGs to the MG database of the mOTUs using vsearch -usearch_global (parameters: --id 0.96 --minqt 0.7). Finally, we evaluated the congruency of the MG-MAGs to mOTU matches. For this, we first checked if at least three MG-MAGs could be assigned to a mOTU (by mapping to a MGC that is part of a mOTU). If this was not the case the MAG was annotated as "unassigned/-1". Next, we removed all alignments to MGCs not assigned to mOTUs and assigned a MAG to a mOTU if >50% of the MG-MAGs were consistently matched to the same mOTU. Otherwise (if no majority mOTU was found) the MAG is annotated as "inconsistent".

**Benchmarking mOTUs2 using simulated metagenomes**. To be able to assess taxonomic quantification accuracy, ten human gut metagenomic samples were simulated using 15,102 Human gut MAGs: a subset of the 19,302 MAGs described before, excluding the MAGs created from samples used to construct the mOTU database (Supplementary Figure 8). MAGs with an ANI > 96.5% were de-replicated to have one representative MAG per species (cut-off according to ref. [4]). The ANI was calculated with the fastANI tool [https://github.com/ParBLiSS/FastANI]. The corresponding fastq files (as well as the simulated abundance data) are available at: http://motu-tool.org/download.html. Metagenomic read data were simulated using BEAR[54]: first, we generated 100 M inserts (2 × 100 M paired-end reads of 150 nt length) with 350 nt insert distance (standard deviation: 30) using generate_reads.py. Second, trim_reads.pl with default parameters was used to add the quality scores, introduce errors and shorten the reads. Every sample was simulated based on mOTUs2 profiled relative abundances from ten real samples. For each simulated sample, we randomly selected 50 MAGs with a representative reference genome sequence in the superset of the Kraken, MetaPhlAn2, or ref-mOTU databases and 50 additional MAGs sampled from those that lacked any reference database representation (which does not preclude these MAGs to map to meta-mOTUs).

The benchmark was performed by evaluating precision-recall plots of the simulated metagenomes based on the number of true positives (TP) false positives (FP) representing species that are predicted but not present in the real sample, and false negatives (FN) representing species that are missed by the profiler. Precision is calculated as TP/(TP + FP) and recall as TP/(TP + FN). Next we evaluated the mean absolute error (MAE) defined as the average absolute difference between estimated relative abundances and relative abundances simulated as ground truth. Finally we evaluated the accuracy of alpha diversity estimates using the difference between predicted and actual Shannon index (abbreviated as $H'$).

**Benchmarking mOTUs2 using the CAMI framework**. We further evaluated mOTUs2 in the CAMI framework[25], which includes eight simulated samples (one low complexity, two medium complexity and five high complexity) for which the ground truth is available. Within the first CAMI community challenge, ten metagenomic profiling tools including MetaPhlAn2 and mOTUs1 were already benchmarked on these data sets. To comparatively assess the performance of mOTUs2 in this context, we converted its output to CAMI/Bioboxes format (-C option in the mOTUs2 profiler) and used OPAL 0.2.9[26] (developed by the same authors as CAMI) for consistency of performance assessments. Using precision-recall plots we evaluated mOTUs2 employing five different parameter sets: high precision (-l 140 -g 6 -C precision), default (-l 100 -g 3 -C precision), recall (-l 75 -g 3 -C recall), high recall (-l 50 -g 2 -C recall) and maximum recall (-l 30 -g 1 -C recall). Hence mOTUs2 are represented by five red dots in the precision-recall plots, demonstrating that it can be tuned to obtain a range of precision-recall trade-offs. The evaluation of the mean absolute error (MAE), which it is called L1 norm in the CAMI paper, was also obtained with OPAL. By default, OPAL re-normalises the relative abundances of the gold standard and the profiling result to each sum to

1 before calculating the MAE, which apparently substantially deteriorates the quantification accuracy of mOTUs2 (see Supplementary Figure 11b). For this reason, we included both re-normalised and relative abundances without any post-processing in our evaluation for mOTUs2. This aims for maximum transparency in the comparison to the other tools, which could only be evaluated with the re-normalised version (but could theoretically also benefit from an evaluation of non-normalised relative abundances).

**Determining environmental specificity of mOTUs**. To determine the environmental specificity of the mOTUs, we used the set of >3100 metagenomes (Supplementary Data 1) to assess the environmental specificity of all meta-mOTUs and the subset of ref-mOTUs that are present in these samples (Supplementary Figure 5a). To this end, we generated mOTUs2 profiles of these samples with default settings and removed samples with less than 500 scaled insert counts. Based on the resulting profiles (https://motu-tool.org/data/All_2481_at_least_500.motu.nr.out.20180307.tsv), we classified a mOTU to be present in a specific environment if it was detected in more than three samples from that environment.

**Analysis of community structure**. We assessed correlations of the Shannon index calculated based on 16S rRNA gene-based analyses and three metagenomic profiling tools (mOTUs2, MetaPhlAn2 and Kraken). For this we used data from two different biomes: metagenomes generated from stool samples of a colorectal cancer (CRC) study[21] and metagenomes from seawater samples of the Tara Oceans expedition[22]. For the CRC study, amplicon sequencing data of the V4 region of the 16S rRNA were downloaded from the European Nucleotide Archive (ENA) database (http://www.ebi.ac.uk/ena): accession number ERP005534. For the ocean water samples, 16S rRNA gene containing fragments were extracted from metagenomic sequencing reads (miTAGs[55]). To ensure comparability between the data sets, we extracted the first 100 bp from each miTAG sequence starting from the V4 primer sequence.

Ribosomal RNA data were initially processed using USEARCH[56] (version 9.2.64) as follows: paired-end reads were merged and quality-filtered using the fastq_mergepairs command with default settings. Merged reads were filtered using the fastq_filter command (-fastq_maxee 0.1). Sequences were de-replicated using the fastx_uniques command, singletons were excluded and the remaining unique sequences were clustered into operational taxonomic units (OTUs) at 97% with chimera removal using the cluster_otus command. Finally, OTU abundances for each sample were determined using the usearch_global command (-strand both; -id 0.97). The OTU abundance tables were downsampled to the minimum number of reads per sample (CRC: 40,805 reads, TARA: 1494 reads) to normalize for uneven sequencing depths using the R function *rarefy* within the *vegan* package[57]. The Shannon index of diversity was computed for each sample and all methods (16S rRNA gene-based and metagenomic method-based) using the R function *diversity* of the *vegan* package. In order to obtain a 95% confidence interval we used bootstrapping (n = 100,000) by resampling pairs of Shannon index values. The confidence intervals reflect the 2.5 and 97.5 percentile of the bootstrapped samples.

Between sample distances were determined using human body site samples for which more than one time point was available for the same individual. More specifically, for each body site, we compared community compositional distances between samples from the same individual (intra-individual) to distances between this and other individuals (inter-individual). Canberra and Bray-Curtis distances were computed with the *vegdist* R function of the *vegan* package and the log-Euclidean distance was computed as the Euclidean distance of the log-transformed relative abundances after the addition of a pseudocount smaller than the smallest non-zero value. For each of the three distances and each sample, we identified the most similar sample (i.e. the one with the minimum distance value) and determined the proportion of cases in which both samples belonged to the same individual.

**Analysis of metatranscriptomes**. To demonstrate the use of mOTUs2 to assess basal transcriptional activity of microbial community members, we used a dataset from 36 samples for which metagenomic and metatranscriptomic sequencing data are available[31]. Each sample (36 metagenomes and 36 metatranscriptomes) was subjected to profiling using mOTUs2, Kraken/Bracken and MetaPhlAn2. All resulting profiles were transformed to relative abundances, and log-transformed after adding a small pseudocount. After that, Spearman correlations between corresponding metagenomic and metatranscriptomic profiles generated from the same sample were calculated and compared between profiling methods (Fig. 4a and Supplementary Figure 11). We moreover evaluated how well species abundance estimates correlated between metagenomic and metatranscriptomic profiles for the twelve most abundant taxa at the class level. Class level information for mOTUs and MetaPhlAn2 was available as part of the profiler output. Class level annotations for Kraken were obtained using NCBI taxonomy identifiers.

**Comparison of SNV profiles from MGs and whole genomes**. To assess the comparability of SNV profiles generated with mOTUs2 and whole genomes, we used samples from 2807 human microbiome samples[14, 15] and 139 prokaryote-enriched metagenomes from the Tara Oceans project[22]. Metagenomic reads were mapped to the mOTUs centroid database using the mOTUs2 command map_snv

and in addition to a set of 5306 reference genomes[13]. Genomic distances of strain populations between samples were estimated based on SNV profiles computed both on mOTUs and the whole genomes using the motus snv_call command. The filtering parameters used within the snv_call command were adapted to the specificity of datasets and references. The allele frequency tables were filtered using a horizontal coverage (-fb) equal to 40% for whole genome-mapped reads and 80% for mOTU-mapped reads, a vertical coverage (-fd) of 10, a per position coverage (-fc) of 5 and a position prevalence (-fp) of 0.90. The minimum number of samples per reference (-fm) was 20 for the human samples and 5 for the Ocean samples. Whole-genome-based distances were compared to those from mOTUs using Pearson's correlation (Fig. 5a). We selected the ref-mOTUs/genomes that passed the filtering thresholds for both methods and correlated between sample distances between the two methods (n.b. there were no species from the vaginal supersite passing the filtering requirements for both methods).

**Individuality of microbial populations across body sites.** We tested for the individuality of microbial strain populations on the subset of the human microbiome samples described above (5.4.1), for which at least two time point data were available. For each body site, we compared SNV profile distances between samples from the same individual (intra-individual, intra-body-site distances) to distances between this and other individuals (inter-individual, intra-body-site distances). To determine whether intra-individual distances were smaller than inter-individual distances (see Supplementary Figure 17b)—indicating individuality of strain populations—we used ROC analysis. ROC curves (see Supplementary Figure 17a) ascertain how accurately small distances predict whether a pair of samples originated from the same individual (with similarly small inter-individual distances being considered false positives) when systematically varying the distance cutoff. ROC curves can be summarized by the area under the curve (AU-ROC) with higher values corresponding to clearer separation between intra- and inter-individual distances (Fig. 5b and Supplementary Figure 17a). Confidence intervals on the AU-ROC (Fig. 5b) were obtained by bootstrapping using the pROC package[58].

**Code availability.** The mOTU profiler version 2 and additional information are available at: https://motu-tool.org. Its source code is accessible at: https://github.com/motu-tool/mOTUs_v2.

**Reporting summary.** Further information on experimental design is available in the Nature Research Reporting Summary linked to this article.

## Data availability

To generate the mOTU database, we used reference genome sequence data from the proGenomes database (http://progenomes.embl.de) as well as metagenomic sequence data from the Genbank Sequence Read Archive (https://www.ncbi.nlm.nih.gov/sra) and the EMBL European Nucleotide Archive (ENA; https://www.ebi.ac.uk/ena) with accession numbers listed in Supplementary Data 1. Human gut metagenomic data and metagenome assembled genomes are available at the ENA (accession numbers are listed in Supplementary Data 2; MAGs can be downloaded from: http://ftp.ebi.ac.uk/pub/databases/metagenomics/mags-gut_19k.tar.gz). The 10 human gut metagenomic samples simulated from metagenome-assembled genomes are available on Zenodo (https://doi.org/10.5281/zenodo.1473645). All other relevant data is available upon request.

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

## Acknowledgements

We thank members of the IT core facility and Y. Yuan for managing computing resources at EMBL, and the IT support team of the Institute of Microbiology and the HPC team at ETH Zürich for managing computing resources and the Euler/Leonhard clusters of ETH Zürich. This work was supported by EMBL, ETH Zürich [PHRT-521], the Helmut Horten Foundation, and the Novartis Foundation for Medical-Biological Research [#17B077], the BMBF through the German Network for Bioinformatics Infrastructure (de.NBI) [#031A537B], and the European Molecular Biology Organization [ALTF 721-2015], LTFCOFUND2013 [PCOFUND-GA-2013-609409].

## Author contributions

S.S., G.Z., D.R.M. and A.M. designed the study and conceived the methodology. A.M., and D.R.M. implemented the method. G.Z., A.M., S.S., D.R.M., R.A., G.S., L.P. implemented and evaluated the benchmarks. H.J.R., M.C., P.H., R.A., P.I.C., L.P.C., T.S.B.S., A.A., A.L.M., R.D.F., J.H.C. provided additional tools or analyses. S.S., G.Z., A.M., D.R.M. wrote the paper with contributions from L.P., G.S., A.A., A.L.M., R.D.F., J.H.C., P.B. and additional input from all other authors.
