## [Peer Review File · Nature Communications]

Reviewers' comments:

Reviewer #1 (Remarks to the Author):

Milanesi, Mende and colleagues describe a novel version of the mOTU tool for the analysis of microbiome samples. While the original mOTU version was built on a more restricted and by now somewhat outdated data source, the new version is built on a large scale database of 100TB sequences of proGenome coupled with meta-assembly based integration of further thousands of species.

mOTUs has shown its usability in numerous studies and the authors evaluate the adequateness of their method on both metagenomics as well as metatranscriptomics datasets, also in comparison to other well established tools such as kraken or metaphlan2. Based on the data shown, the authors claim better performance. The statistical analysis is up to the standard of the field and reproducibility is ensured by the provision of a conda environment.

Overall, I have the following concerns:

1) CAMI (<https://www.nature.com/articles/nmeth.4458>) established standardized datasets to facilitate comparisons of metagenomics tools and serves as a gold standard, which should also be included in this evaluation.

This holds in particular since CAMI showed that mOTUv1 was not among the best performing tools and kraken or Metaphlan2 outperformed mOTUv1 in many cases. Rather than comparing mOTUv2 only on their own data, the authors should refer to the established commuting standard and show that mOTUv2 indeed outperforms kraken and Metaphlan2 on independent data. This would also allow estimating the benefit of switching from mOTUv1 to mOTUv2 in direct comparison.

2) With mOTUv1, it became rather evident that its database was not up to date and many organisms characterized by other approaches based on the newest refseq version, were missed. While mOTUv2 fixes this problem for now, I do wonder if it was not time to provide an automated update of the data source, too or at least the scripts for running it. This is standard for practically any other current metagenomics tool and making a tool dependent on a static data source does not reflect the strong growth of databases, which actually motivates this contribution.

3) The comparison with kraken needs to be on the same database with kraken. Otherwise it is comparing apples and oranges and it has been often demonstrated that the impact of the database is far more influential than the question of the computational tool. A recent preprint clearly shows the database dependence of kraken (<https://doi.org/10.1101/304972>). Currently the authors rely on a 8GB version of minikraken but base their own tool proGenomes which is based on 100GBp NCBI bacterial genomes plus many metagenome samples. To allow fair comparison of tools, the authors need to build a comparable custom database for kraken, otherwise the tool comparison is of minimal meaning.

4) It has been shown that merging results from motu with whole sequence based identification such as kraken or clark significantly improves results (<https://doi.org/10.1186/s40168-017-0318-y>). It would be insightful to investigate whether this holds for mOTUv2.

5) The data used for creating mOTUv2 Supplementary table1 is then partially used for validating mOTUv2 in comparison to other approaches (Supplementary table2).

It would clearly be preferable to have non-overlapping datasets otherwise it may be a self-fulfilling hypothesis.

Reviewer #2 (Remarks to the Author):

The manuscript "Microbial abundance, activity, and population genomic profiling with mOTUs" by Milanese et al. describes an updated method for the identification of metagenomic-based operational taxonomic units (mOTUs) that extend previous work from the same group. Application of the method to a set of >3100 metagenomic samples allowed the compilation of a mOTU database that includes >7500 mOTU profiles, 2/3 of which correspond to known species. A metagenomic taxonomic profiler using this dataset of profiles provides high accuracy. Comparing the estimates of sample diversity to what obtained using assembly-based methods, the authors note that while all assembled genomes can be assigned to a mOTU, only 42% of the mOTUs can be associated to one assembled genome. The authors also show useful applications of this approach, including the use of Single Nucleotide Variant profiles to identify strains in metagenomic samples. Noting that there was a good correlation between the relative abundances between mOTU profiles of matched metagenomic and metatranscriptomic experiments (although in a relatively small dataset, only 36 samples), the authors conclude that the ten genes used for mOTU profiling are constitutively expressed, thus providing a reliable baseline for metatranscriptomics experiments.

The paper is clear and well written. However, the mOTUs and the underlying characterization of the universal, single-copy marker genes have been already presented in at least two high profile papers (Mende, D.R., et al., Accurate and universal delineation of prokaryotic species. *Nat Methods*, 2013. 10(9): p. 881-4; Sunagawa, S., et al., Metagenomic species profiling using universal phylogenetic marker genes. *Nat Methods*, 2013. 10(12): p. 1196-9), where the choice of the marker genes, the methods for mOTUs definition, the use of mOTUs to taxonomically profile metagenomic samples, the accuracy of the method and the major biological insights that can be obtained using mOTUs have already been demonstrated. It is not clear what major methodological and/or biological advancement has been achieved here.

Specific points:

I apologize if I cannot clearly indicate the points I am referring to, but my copy of the manuscript lacks line numbers.

- On page 4 see also sup. Figure 5, the authors note that "A breakdown of mOTUs by biome showed that ref-mOTUs are often detected in multiple biomes, while meta-mOTUs tend to be more biome-specific (SFig. 5a)". The authors comment in the caption of Supp. Figure 5 that "The ref-mOTUs appear to be shared between more biomes, reflecting the fact that are easier to cultivate. On the other hand, meta-mOTUs capture species that are biome-specific." There is another explanation, namely the possibility that different species are highly correlated in homogeneous samples (i.e. same biome) but not in inhomogeneous samples (i.e. different biomes), thus leading to the definition of spurious entities when the different MGs are connected in the co-abundance based binning step. These spurious mOTUs would be more biome specific, as found by the authors. The authors should comment on this.
- On page 6, figure 1d, the authors benchmark mOTUs against Kraken and MetaPhlAn2 using synthetic datasets composed by simulated metagenomics samples containing 50 MAGs with and 50 MAGs without a representative reference genome sequence. As MetaPhlAn2 (and I believe also Kraken) relies on reference genomes, I don't think that this is a fair comparison, because MetaPhlAn2 will, by definition, miss all MAGs that do not have a reference genome.
- Figure 2. I don't think it is a useful piece of information for the reader that mOTUs gives a better

estimate of a sample alpha diversity than MetaPhlan2 and Kraken, since these two tools have a different purpose. The reference method in this case is the use of 16S sequences. I would suggest the authors to show a comparison of the mOTU method with a 16S-based method both on samples of known (simulated, mock communities if available) and unknown composition.

Reviewer #3 (Remarks to the Author):

Review for "Microbial abundance, activity, and population genomic profiling with mOTUs"

The work by Milanese et al. presents an updated version of the mOTU metagenome profiler for computing relative microbial abundances at the species level of input metagenome reads. The method relies on the alignment of the reads to a database of single copy marker genes, which are clustered into so-called mOTUs (phylogenetic marker gene-based operational taxonomic units) denoting each a different species. As in the previous version, the method enables the estimation of the relative abundances of microbes even for those without representative sequences in reference databases. This is done by aggregating in the database marker genes extracted from both reference databases and metagenome assembled genomes.

A major change in this new version of mOTU is the updated database increasing 7 times the number of unknown species that can be profiled compared to its previous version, in particular of species associated with different human body sites and the ocean. Another contribution is the capability to quantify transcriptional activity of microbial community members from metatranscriptomes, showing on available data that transcript abundances computed from metagenomes and metatranscriptomes correlate more strongly for mOTU than when computed with other methods (Kraken and MetaPhlan2). In addition, the manuscript shows that the marker genes suitable for application in metagenomics and comprising the mOTUs can be used to compute profiles of single nucleotide variations. This is a fast alternative to computing profiles from whole genomes and facilitates the investigation of microbial communities on the strain level.

To assess the profiling performance of the method, the authors simulated samples of reads from metagenome assembled genomes from the human gut and marine environments, for each sample using metagenome assembled genomes with and without a representative reference genome sequence. mOTU outperformed the compared methods Kraken and MetaPhlan2 by a large margin in terms of predicted relative species abundances, number of false negatives, and estimation of alpha and beta diversities.

The substantial increase in the number of unknown species that can be identified in metagenome data and the added capabilities make this work very relevant to the field. The software and used data are available online and are well documented.

Major

-Despite the sensible experiment dataset, the methods could also have been assessed on already available datasets, such as the one used in the MetaPhlan2 manuscript (Truong et al. 2015), which shows MetaPhlan2 outperforming mOTU version 1.

-For a broader comparison to other programs, assessing mOTU on the benchmark data of the Critical Assessment of Metagenome Interpretation (CAMI, Sczyrba et al. 2017) would be very interesting. This would also indicate how much mOTU improves profiling of metagenome data from environments other

than human and marine. This comparison would be particularly meaningful if the genome resource used for MG identification did not include the genomes used in the CAMI challenge (this would be the case if only genomes from proGenomes (Mende et al. 2017) that were released before mid 2015). The results of other tools on some CAMI datasets are available together with the OPAL benchmarking package: <https://www.biorxiv.org/content/early/2018/07/19/372680>.

-Although claiming improved sensitivity, it is actually not computed in the comparison between mOTU and the competing methods (except for absolute values in Fig. 1d).

-To further justify the need for a new version of mOTU, the authors should include in their comparisons the previous version of mOTU.

-To facilitate the use of mOTU, the authors could provide outputs in more common formats, such as the BIOM (McDonald et al. 2012) or CAMI profiling output (<https://github.com/bioboxes/rfc/blob/master/data-format/profiling.mkd>) formats.

Minor

End of page 9: "we found overall that the distances of SNV profiles using MGs correlated were highly correlated".

Double use of word "correlated".

Beginning of page 20: "These four steps are directly build upon metaSNV capabilities".
build -> built

Why does SFig. 12 have a yellow circle over some plots?

Reviewer #1 (Remarks to the Author):

Milanese, Mende and colleagues describe a novel version of the mOTU tool for the analysis of microbiome samples. While the original mOTU version was built on a more restricted and by now somewhat outdated data source, the new version is built on a large scale database of 100TB sequences of proGenome coupled with meta-assembly based integration of further thousands of species.

mOTUs has shown its usability in numerous studies and the authors evaluate the adequateness of their method on both metagenomics as well as metatranscriptomics datasets, also in comparison to other well established tools such as kraken or metaphlan2. Based on the data shown, the authors claim better performance. The statistical analysis is up to the standard of the field and reproducibility is ensured by the provision of a conda environment.

Overall, I have the following concerns:

1) CAMI (<https://www.nature.com/articles/nmeth.4458>) established standardized datasets to facilitate comparisons of metagenomics tools and serves as a gold standard, which should also be included in this evaluation.

This holds in particular since CAMI showed that mOTUv1 was not among the best performing tools and kraken or MetaPhlan2 outperformed mOTUv1 in many cases. Rather than comparing mOTUs2 only on their own data, the authors should refer to the established commuting standard and show that mOTU2 indeed outperforms kraken and MetaPhlan2 on independent data. This would also allow estimating the benefit of switching from mOTUv1 to mOTUv2 in direct comparison.

Response:

We would like to thank the reviewer for the excellent and constructive suggestions.

We would like to note that originally, we had not included a benchmark using the CAMI dataset:

- (i) due to difficulties in mapping the mOTU taxonomic annotations to the NCBI taxonomy on which the CAMI evaluations are based in particular at the species level, which is central to the mOTU clustering approach (at this high taxonomic resolution, meta-mOTUs without available reference sequences cannot be evaluated at all),*
- (ii) due to taxonomic uncertainties in the CAMI gold standard, again particularly at the species level, and*

(iii) due to limited availability of resources that were needed to evaluate mOTUs2 in the context of the CAMI framework.

However, after revising these issues, we could finally establish a mapping to the CAMI taxonomy, which allowed us to benchmark mOTUs2 using the OPAL package. Our results now show that mOTUs2 outperformed most other tools in many evaluation metrics, in particular for simulated metagenomes of medium and high complexity. More specifically, (i) mOTUs2 outperformed mOTUs1 at all taxonomic ranks, (ii) and also other tools (including MetaPhlan2; note that Kraken/Bracken was not tested in CAMI) at the genus and species level (see Figure 2k-o; and Suppl. Fig. 10-11).

Taken together, we believe that we have addressed the reviewer's concern. As the new benchmarking results have added so much value to the manuscript, we present them now in an additional figure.

Actions taken:

- The mOTU profiler now has a new option ("-C") that provides the output in CAMI format.
- We used the new CAMI profiles to benchmark mOTUs, both version 1 and version 2 (as suggested), against other tools using the simulated metagenomes provided by CAMI. The results are provided, along with our own improved benchmarks (see point 5), in Figure 2, and discussed in the main text.

2) With mOTUv1, it became rather evident that its database was not up to date and many organisms characterized by other approaches based on the newest refseq version, were missed. While mOTUv2 fixes this problem for now, I do wonder if it was not time to provide an automated update of the data source, too or at least the scripts for running it. This is standard for practically any other current metagenomics tool and making a tool dependent on a static data source does not reflect the strong growth of databases, which actually motivates this contribution.

Response:

We fully understand the reviewer's concern regarding the lack of a procedure that would allow for an automated update of the data resource, although we respectfully disagree with the statement that this is standard for any other current metagenomics tools. For example, MetaPhlan2, FOCUS, Taxy-Pro, TIPP (all evaluated by the CAMI challenge) do not provide this possibility. For Kraken/Bracken, the difficulty to construct the database is documented in the preprint that was provided by the reviewer (see point 3).

The reason why mOTUs cannot be easily updated is that a number of manual curation and parameter optimization steps were required to establish the final database. These included the identification and removal of outliers, determining biome-specific predictive strength cutoffs for the binning of marker gene clusters into mOTUs, manual curation of the last common ancestor-based taxonomic annotation and consolidation of mOTUs with NCBI taxonomy at the species and to a lesser extent also other taxonomic levels.

Nevertheless, we are planning to provide more frequent updates in the future, and keep working on automating as many steps as possible. As a first step towards this goal, we have published a detailed tutorial and the software tools required to enable expert users to add reference genomes, SAGs and/or MAGs to the default databases. However, we would like to emphasize the need for manual quality control, taxonomic annotation and curation.

Action taken:

- We now provide a detailed tutorial for expert users to add new reference genomes, SAGs or MAGs to the mOTU database: <http://motu-tool.org/installation.html>

3) The comparison with kraken needs to be on the same database with kraken. Otherwise it is comparing apples and oranges and it has been often demonstrated that the impact of the database is far more influential than the question of the computational tool. A recent preprint clearly shows the database dependence of kraken (<https://doi.org/10.1101/304972>). Currently the authors rely on a 8GB version of minikraken but base their own tool proGenomes which is based on 100GBp NCBI bacterial genomes plus many metagenome samples. To allow fair comparison of tools, the authors need to build a comparable custom database for kraken, otherwise the tool comparison is of minimal meaning.

Response:

We understand the reviewer's concern, based on a preprint article, that differences in the reference database used by taxonomic profiling tools may impact results. However, updating the Kraken database poses a serious challenge even to expert bioinformaticians: According to the mentioned preprint, rebuilding a Kraken database from version 70 of refseq took 500GB RAM and 2 days/64 cores, from version 80 2.5TB RAM, 11 days/64 cores, and future releases will need more than 4TB RAM and weeks of computation. Database updates get even more involved when the Bracken database has to be updated as well. In cases like this, where tool optimisation is so technically challenging and time-consuming that the vast majority of users would not attempt it, we feel it is fair to evaluate the tools as they are made available to the community. This is also accepted practice in community challenges such as CAMI (unless the developers are actively participating and making their optimisations available). These evaluations address the average user, who would in most cases use the provided database that according to the Kraken website has 99% genus precision and 91% genus sensitivity.

Also, regarding the database size, even with metagenomic data included, the mOTUs2 profiler only uses 600 Mb of marker gene sequences as a reference database. Thus, we do not feel that database size necessarily provides an advantage. This is also one of the findings of the cited preprint paper.

More importantly, we now used the CAMI standardized dataset, as suggested by the reviewer, for broader comparisons to other metagenomic profiling tools. Unfortunately, the original CAMI challenge did not evaluate Kraken/Bracken. However, as an alternative, we compared our results for mOTUs2 and Kraken/Bracken using the simulated metagenome provided by the developers of MetaPhlan2. In this benchmark our Kraken/Bracken results are very similar to, or even better than (smaller error for even dataset), those generated by the MetaPhlan2 authors for Kraken originally; Kraken/Bracken performs still worse than mOTUs2 (Suppl. Table 5).

Overall, we appreciate the reviewers remark that differences in the reference databases used by different tools can impact the results. However, we feel that the way we evaluate the performance of different tools, which is the primary goal, is -- also thanks to the suggestion made by the reviewers -- greatly expanded in the revised version of our manuscript and now at least as thorough as is standard for other taxonomic profiler publications we are aware of.

Action taken:

- We included an additional benchmark using the simulated metagenomes provided by the MetaPhlan2 authors finding that our use of Kraken/Bracken yields very similar (or better) results than Kraken as originally evaluated in Truong et al., Nat Methods, 2015.

4) It has been shown that merging results from motu with whole sequence based identification such as kraken or clark significantly improves results (<https://doi.org/10.1186/s40168-017-0318-y>). It would be insightful to investigate whether this holds for mOTUv2.

Response:

We would like to thank the reviewer for pointing out that merging results from different tools may improve taxonomic profiling results. The provided reference describes a tool named MetaMeta (Piro et al., Microbiome, 2017), that merges results from different taxonomic profiling tools, using datasets provided by CAMI and real metagenomes from the Human Microbiome Project (HMP). To test how mOTUs2 compares to such ensemble approaches, we profiled the same benchmark dataset with different options and find that mOTUs2 as a stand alone tool outperformed MetaMeta (RFigure1), which had not been the case for mOTUsv1. We also note that in Fig. 6 of the MetaMeta paper, which is based on real metagenomes from the HMP, mOTUs1 was shown to perform significantly better (about an order of magnitude lower false positive rates at a similar true positive rate) than MetaMeta.

While we do not know whether other tools would like to use mOTUs2, we provide the prerequisites that would allow for a seamless integration if desired (Biobox / BIOM/CAMI output format, docker containers, and availability through bioconda).

Action taken:

- We tested mOTUs2 against MetaMeta (the ensemble tool referenced by the reviewer) using the same benchmark dataset as in the reference and find it to perform better than merged results. The results are shown below:

RFigure 1: Precision-Recall benchmark against MetaMeta for CAMI simulated metagenomes of high (top), medium (middle) and low (bottom) complexity. In all comparisons, the mOTUs2 tool alone significantly outperforms the ensemble approach of merging profiling results as implemented in MetaMeta (Piro et al., Microbiome, 2017).

5) The data used for creating mOTUv2 Supplementary table1 is then partially used for validating mOTUv2 in comparison to other approaches (Supplementary table2). It would clearly be preferable to have non-overlapping datasets otherwise it may be a self-fulfilling hypothesis.

Response:

We would like to thank the reviewer for noting this shortcoming of our previous dataset for benchmarking. We followed the suggestion and re-ran the simulations after excluding the samples that were used for the creation of the database. Although this did not affect any of the conclusions, we feel that the procedure is now 'cleaner'. Details are now described by the addition of a new supplementary figure.

Actions taken:

- *To improve our benchmark, we re-simulated MAG communities and excluded samples present in the database used to generate mOTUs.*
- *Added Supplementary Fig. 8 to describe the benchmarking procedure in detail.*

Reviewer #2 (Remarks to the Author):

The manuscript "Microbial abundance, activity, and population genomic profiling with mOTUs" by Milanese et al. describes an updated method for the identification of metagenomic-based operational taxonomic units (mOTUs) that extend previous work from the same group. Application of the method to a set of >3100 metagenomic samples allowed the compilation of a mOTU database that includes >7500 mOTU profiles, 2/3 of which correspond to known species. A metagenomic taxonomic profiler using this dataset of profiles provides high accuracy. Comparing the estimates of sample diversity to what obtained using assembly-based methods, the authors note that while all assembled genomes can be assigned to a mOTU, only 42% of the mOTUs can be associated to one assembled genome. The authors also show useful applications of this approach, including the use of Single Nucleotide Variant profiles to identify strains in metagenomic samples. Noting that there was a good correlation between the relative abundances between mOTU profiles of matched metagenomic and metatranscriptomic experiments (although in a relatively small dataset, only 36 samples), the authors conclude that the ten genes used for mOTU profiling are constitutively expressed, thus providing a reliable baseline for metatranscriptomics experiments.

The paper is clear and well written. However, the mOTUs and the underlying characterization of the universal, single-copy marker genes have been already presented in at least two high profile papers (Mende, D.R., et al., Accurate and universal delineation of prokaryotic species. Nat Methods, 2013. 10(9): p. 881-4; Sunagawa, S., et al., Metagenomic species profiling using universal phylogenetic marker genes. Nat Methods, 2013. 10(12): p. 1196-9), where the choice of the marker genes, the methods for mOTUs definition, the use of mOTUs to taxonomically profile metagenomic samples, the accuracy of the method and the major biological insights that can be obtained using mOTUs have already been demonstrated. It is not clear what major methodological and/or biological advancement has been achieved here.

Response:

We would like to thank the reviewer for pointing out that the methodological advancement and the biological analyses that these enable were not sufficiently clear. It is correct that the choice of marker genes, the definition of mOTUs, the concept of using mOTUs for taxonomic profiling of metagenomes had been described in our earlier work (Mende et al., 2013; Sunagawa et al., 2013). However, the new version encompasses, in addition to an update in the sequence database, methodological improvements, namely, a more precise method for co-abundance based binning able to work across samples from heterogeneous environments, and two major functional novelties, which had previously not been described nor implemented or benchmarked: the capacity to profile metatranscriptomes for basal transcriptional activity and to profile single nucleotide variations between strain populations. The latter now provides the possibility to profile microbial communities beyond the species level, the use of which we exemplify in our analysis of strain retention in different human body parts. To better highlight these novel aspects, we revised the abstract of our manuscript.

Action taken:

- *We restructured and rephrased parts of the abstract to emphasize more clearly the novel functionalities of the mOTU profiler. The text now reads:
"We present **an updated and functionally extended tool** [...]"*

and specifically, we now state:

“As a new feature, we show that mOTUs, which are based on genes for essential housekeeping functions, are demonstrably well-suited for quantification of basal transcriptional activity of community members. Furthermore, single nucleotide variation profiles estimated using mOTUs reflect those from whole genomes, which allows for comparing, e.g., differences and individuality of microbial strain populations across different human body sites.”

Specific points:

I apologize if I cannot clearly indicate the points I am referring to, but my copy of the manuscript lacks line numbers.

- On page 4 see also sup. Figure 5, the authors note that “A breakdown of mOTUs by biome showed that ref-mOTUs are often detected in multiple biomes, while meta-mOTUs tend to be more biome-specific (SFig. 5a)”. The authors comment in the caption of Supp. Figure 5 that “The ref-mOTUs appear to be shared between more biomes, reflecting the fact that are easier to cultivate. On the other hand, meta-mOTUs capture species that are biome-specific.” There is another explanation, namely the possibility that different species are highly correlated in homogeneous samples (i.e. same biome) but not in inhomogeneous samples (i.e. different biomes), thus leading to the definition of spurious entities when the different MGs are connected in the co-abundance based binning step. These spurious mOTUs would be more biome specific, as found by the authors. The authors should comment on this.

Response:

We thank the reviewer for mentioning an important potential confounder, that may challenge our statement that meta-mOTUs tend to be more biome-specific than ref-mOTUs. In fact, we had been concerned about the same point, which is why we benchmarked the binning based on reference genomes (SFig. 2). However, we now included an additional test using an individual MG (COG0012) to derive the same estimates, but independently of binning, and find that the trend of higher biome-specificity for meta-mOTUs does not change (SFig. 5b).

Action taken:

- *To rule out potential binning artefacts, we included an additional analysis of biome-specificity of meta-mOTUs compared to ref-mOTUs by evaluating only a single MG, that is independent of binning. The results have been incorporated into SFig. 5.*

- On page 6, figure 1d, the authors benchmark mOTUs against Kraken and MetaPhlAn2 using synthetic datasets composed by simulated metagenomics samples containing 50 MAGs with and 50 MAGs without a representative reference genome sequence. As MetaPhlAn2 (and I believe also Kraken) relies on reference genomes, I don't think that this is a fair comparison, because MetaPhlAn2 will, by definition, miss all MAGs that do not have a reference genome.

Response:

The distinct feature of mOTUs is to account for species that currently lack representative reference genomes, which is evident even for well-studied communities such as the gut microbiome (see e.g. Karst et al., Nat Biotechnol., 2018 for an independent report). This is indeed a critical advantage of the approach that highlights a blind spot of many other tools for taxonomic profiling of metagenomes. It is thus our intention to clearly describe this advantage in our manuscript and to show its effects on the accuracy of taxonomic profiling compared to other tools through benchmarks that are able to address this.

To allow readers to compare these benchmarks to previous ones that were exclusively based on sequenced isolate genomes, we now additionally used (i) the simulated metagenome provided by the developers of MetaPhlAn2 and (ii) the datasets provided by CAMI (<https://www.nature.com/articles/nmeth.4458>) that were provided to the community with the specific goal of enabling fair benchmarks, including for taxonomic profiling. As a result, we find that mOTUs2 performs similarly or better than other tools and included these results in the revised version of the manuscript.

Action taken:

- *We used the simulated metagenomes provided by the developers of MetaPhlAn2, one of the tools against which mOTUs2 is compared to. In addition, we performed an extensive comparison of mOTUs2 with other tools using the CAMI benchmark data set.*
- *New benchmarking results are shown in Fig. 2, Suppl. Fig. 9-11, and Suppl. Tables 3-5.*
- *We further emphasized the importance of quantifying species for which reference genomes are currently unavailable. The text reads now:
"more than 50% of gut microbial species still lack representative reference genomes (see also³⁸), which may seem unexpected, but this estimate is in the same range as reported for an independent approach³⁹"*

• Figure 2. I don't think it is a useful piece of information for the reader that mOTUs gives a better estimate of a sample alpha diversity than MetaPhlAn2 and Kraken, since these two tools have a different purpose. The reference method in this case is the use of 16S sequences. I would suggest the authors to show a comparison of the mOTU method with a 16S-based method both on samples of known (simulated, mock communities if available) and unknown composition.

Response:

We appreciate the reviewer's concern about the purpose of MetaPhlAn2 and Kraken being different from the one of mOTUs. However, we respectfully disagree on this point, as the purpose of any taxonomic profiling tool is to identify and estimate the relative abundance of ideally all taxa present in a microbial community. Alpha diversity is a function of the richness (i.e., the number of species) and the frequency (relative abundance) of all taxa, and clearly many previous shotgun metagenomic analyses were based on the alpha diversity estimates derived from the profiling tools we compare mOTUs2 to (including in the HMP, see Huttenhower et al. Nature 2012, Fig. 1; see for further examples also Ananthkrishnan et al., Cell Host Microbe, 2017; Ai et al., BMC Genomics, 2017).

We do agree with the reviewer that 16S-based methods are often used for estimating alpha diversity. Part of the reason why reference genome-based tools provide less accurate estimates is due to the unclassified fraction of taxa (see also Supplementary Fig. 1), which is (partly) accounted for by mOTUs. Based on this, we suggest that mOTUs will provide a better estimate of alpha diversity. To test this, we did in fact, as suggested by the reviewer, use 16S-based data to compare mOTUs and other tools on samples of unknown composition (now Figure 3); for simulated data we have directly compared diversity estimates to the ground truth (Figure 2g).

Action taken:

- *We included an additional evaluation of alpha diversity estimates based on simulated communities (Figure 2g)*

Reviewer #3 (Remarks to the Author):

Review for "Microbial abundance, activity, and population genomic profiling with mOTUs"

The work by Milanese et al. presents an updated version of the mOTU metagenome profiler for computing relative microbial abundances at the species level of input metagenome reads. The method relies on the alignment of the reads to a database of single copy marker genes, which are clustered into so-called mOTUs (phylogenetic marker gene-based operational taxonomic units) denoting each a different species. As in the previous version, the method enables the estimation of the relative abundances of microbes even for those without representative sequences in reference databases. This is done by aggregating in the database marker genes extracted from both reference databases and metagenome assembled genomes.

A major change in this new version of mOTU is the updated database increasing 7 times the number of unknown species that can be profiled compared to its previous version, in particular of species associated with different human body sites and the ocean. Another contribution is the capability to quantify transcriptional activity of microbial community members from metatranscriptomes, showing on available data that transcript abundances computed from metagenomes and metatranscriptomes correlate more strongly for mOTU than when computed with other methods (Kraken and MetaPhlAn2). In addition, the manuscript shows that the marker genes suitable for application in metagenomics and comprising the mOTUs can be used to compute profiles of single nucleotide variations. This is a fast alternative to computing profiles from whole genomes and facilitates the investigation of microbial communities on the strain level.

To assess the profiling performance of the method, the authors simulated samples of reads from metagenome assembled genomes from the human gut and marine environments, for each sample using metagenome assembled genomes with and without a representative reference genome sequence. mOTU outperformed the compared methods Kraken and MetaPhlan2 by a large margin in terms of predicted relative species abundances, number of false negatives, and estimation of alpha and beta diversities.

The substantial increase in the number of unknown species that can be identified in metagenome data and the added capabilities make this work very relevant to the field. The software and used data are available online and are well documented.

Major

-Despite the sensible experiment dataset, the methods could also have been assessed on already available datasets, such as the one used in the MetaPhlan2 manuscript (Truong et al. 2015), which shows MetaPhlan2 outperforming mOTU version 1.

Response:

We would like to thank the reviewer for the suggestion to assess the performance of mOTUs2 using the available MetaPhlan2 dataset. In this additional evaluation, we find that the mOTUs2 profiler performs comparably to MetaPhlan2 and better than mOTUs1, Kraken (as tested by the original authors), and Kraken/Bracken (in a similar setting to those in the other evaluations of our manuscript).

Action taken:

- *We followed the reviewer's suggestion to use the simulated metagenomes of the MetaPhlan2 manuscript and compare the accuracy of mOTUs2 to those of MetaPhlan2, mOTUs1, Kraken/Bracken. The results are presented in Supplementary Table 5 and mentioned in the main text as: "Finally, since Kraken was not included in the CAMI benchmark²⁵ dataset, we compared the performance of mOTUs2 to the results reported for the evaluation of MetaPhlan2⁷, which included Kraken⁶. We find that mOTUs2 and MetaPhlan2 performed similarly, while both (and mOTUs1) outperformed Kraken (Supplementary Table 5)."*

-For a broader comparison to other programs, assessing mOTU on the benchmark data of the Critical Assessment of Metagenome Interpretation (CAMI, Sczyrba et al. 2017) would be very interesting. This would also indicate how much mOTU improves profiling of metagenome data from environments other than human and marine. This comparison would be particularly meaningful if the genome resource used for MG identification did not include the genomes used in the CAMI challenge (this would be the case if only genomes from proGenomes (Mende et al. 2017) that were released before mid 2015). The results of other tools on some CAMI datasets are available together with the OPAL benchmarking package: <https://www.biorxiv.org/content/early/2018/07/19/372680>.

Response:

We would like to thank the reviewer for the excellent and constructive suggestion. (As this has also been raised by Reviewer #1, we duplicate our response here.)

We would like to note that originally, we had not included a benchmark using the CAMI dataset:

(i) due to difficulties in mapping the mOTU taxonomic annotations to the NCBI taxonomy on which the CAMI evaluations are based in particular at the species level, which is central to the mOTU clustering approach (at this high taxonomic resolution, meta-mOTUs without available reference sequences cannot be evaluated at all), (ii) due to taxonomic uncertainties in the CAMI gold standard, again particularly at the species level, and (iii) due to limited transparency and availability of resources that were needed to evaluate mOTUs2 in the context of the CAMI framework.

However, after revising these issues, we could finally establish a mapping to the CAMI taxonomy, which allowed us to benchmark mOTUs2 using the OPAL package. Our results now show that mOTUs2 outperformed most other tools in many evaluation metrics, in particular for simulated metagenomes of medium and high complexity. More specifically, (i) mOTUs2 outperformed mOTUs1 at all taxonomic ranks, (ii) and also other tools (including

MetaPhlan2; note that Kraken/Bracken was not tested in CAMI) at the genus and species level (see Figure 2k-o; and Suppl. Fig. 10-11).

Taken together, we believe that we have addressed the reviewer's concern. As the new benchmarking results have added so much value to the manuscript, we present them now in an additional figure.

Actions taken:

- The mOTU profiler now has a new option ("-C") that provides the output in CAMI format.
- We used the new CAMI profiles to benchmark mOTUs, both version 1 and version 2 (as suggested), against other tools using the simulated metagenomes provided by CAMI. The results are provided, along with our own improved benchmarks (see point 5), in Figure 2, and discussed in the main text.

-Although claiming improved sensitivity, it is actually not computed in the comparison between mOTU and the competing methods (except for absolute values in Fig. 1d).

Response:

We would like to thank the reviewer for this constructive remark. We now show precision-recall plots throughout our benchmarking results.

Action taken:

- Sensitivity (i.e. recall) and precision has been computed for the comparison between mOTUs2 and other tools. The results are shown in Figure 2 and SFig. 10.

-To further justify the need for a new version of mOTU, the authors should include in their comparisons the previous version of mOTU.

Response:

This is also an excellent suggestion. We have now included a thorough comparison to the previous version (mOTUs1).

Action taken:

- mOTUs1 results have been incorporated into our benchmarks (Figure 2, SFig. 10, SFig. 11, and Suppl Table 3-4-5).

-To facilitate the use of mOTU, the authors could provide outputs in more common formats, such as the BIOM (McDonald et al. 2012) or CAMI profiling output (<https://github.com/bioboxes/rfc/blob/master/data-format/profiling.mkd>) formats.

Response:

BIOM and CAMI formats are now supported.

Action taken:

- BIOM and CAMI output formats are now supported using the options (-B) and (-C), respectively.

Minor

End of page 9: "we found overall that the distances of SNV profiles using MGs correlated were highly correlated". Double use of word "correlated".

Response:

This has been corrected.

Beginning of page 20: "These four steps are directly build upon metaSNV capabilities".
build -> built

Response:

This has been corrected.

Why does SFig. 12 have a yellow circle over some plots?

Response:

This was not intentional and has been corrected.

Reviewers' comments:

Reviewer #1 (Remarks to the Author):

The authors undertook great effort to reply to the issues that I raised, however, several of the main aspects remain unsolved in this revised version.

My criticism centers mostly around the issue raised in aspects (2) and (3), but also affects issue (1) and (4). Currently I only regard issue (5) as fully resolved.

The heart of the problem is as follows: Metagenomic reference data is growing exponentially: e.g. Refseq microbial has increased more than twenty-fold within the last four years. When comparing to tools on data from different dates, will mostly reveal the tool as performing best that uses the most novel data source and not the algorithmically best solution. In order to provide any meaningful comparison the authors need to base all tool comparisons on *current* data, not on historic ones. The authors argue that users in practice will not rebuilt databases (which is partially true), but this is not good practice and should not be encouraged.

This affects the following issues:

(1) I truly appreciate the efforts by the authors to compare on CAMI data. As far as I understand from the description in the manuscript, the authors ran their analysis with recent data, but did not rerun the other analyses with most recent data – thus any difference in performance could be due to two reasons: (i) database size or (ii) conceptual advances of mOTUv2. Given the current setup these effects cannot be disentangled and no conclusion can be drawn.

(2) I appreciate the plans by the authors to update mOTU v2 regularly. This commitment should be explicitly stated in the manuscript: 'The authors plan to provide X updates for mOTU v2 per year'. The authors and I appear to agree that without these updates, mOTU v2 would soon be outdated.

(3) I understand the difficulties of the authors to build a kraken database. While the limitations described by the authors do appear solvable to me (I happen to know that Zurich for instance has an excellent support in scientific computations that should not have trouble providing such a machine which could also be easily be rented in the cloud). Still, there are several publications (centrifuge, krakenHII, ganon) that show improved variants of the kraken approach that more easily cope with larger databases. If infrastructural problems do persist that may be a sound alternative to show that the reason for better performance is not solely based on larger databases.

(4) Again, I appreciate the efforts by the authors, but my original questions whether integration with other tool results improves on mOTU v2 remains unanswered. Further, the comparison would be most useful if carried out on most recent database versions.

Reviewer #2 (Remarks to the Author):

The authors have addressed all my concerns

Reviewer #3 (Remarks to the Author):

The authors have done a great job addressing all comments, I have not further suggestions.

Reviewer #1 (Remarks to the Author):

The authors undertook great effort to reply to the issues that I raised, however, several of the main aspects remain unsolved in this revised version.

My criticism centers mostly around the issue raised in aspects (2) and (3), but also affects issue (1) and (4). Currently I only regard issue (5) as fully resolved.

The heart of the problem is as follows: Metagenomic reference data is growing exponentially: e.g. Refseq microbial has increased more than twenty-fold within the last four years.

When comparing to tools on data from different dates, will mostly reveal the tool as performing best that uses the most novel data source and not the algorithmically best solution. In order to provide any meaningful comparison the authors need to base all tool comparisons on *current* data, not on historic ones. The authors argue that users in practice will not rebuilt databases (which is partially true), but this is not good practice and should not be encouraged.

Response:

We acknowledge the reviewers additional concerns, but respectfully disagree with the assessment of what is required for a meaningful comparison of metagenomic profiling tools. We are also surprised that the additional requests are partly not contextualized with the new analyses performed and discussions provided in our previous point-by-point responses.

Regarding the major point raised by the reviewer, the whole field recognizes that the comparison of metagenomic profiling tools is a technically challenging undertaking. In particular, there had been a lack of consensus about benchmarking preventing a fair assessment of tool performance. To address this issue, The Critical Assessment of Metagenome Interpretation (CAMI) challenge has engaged the global developer community to establish a standardized procedure for tool comparison (Sczyrba et al., Nat Methods, 2017). Notably, in the first round of revision, the reviewer appeared to share this view, and specifically asked us to use the CAMI "standardized datasets to facilitate comparisons of metagenomic tools", for which they serve "as a gold standard, which should also be included in this [i.e. our] evaluation". We were, and still are, very grateful for this constructive suggestion as it revealed that mOTUs2 outperform most other tools as presented in the revised version of the manuscript (Figure 2k-o; and Suppl. Fig. 10-11). However, despite fully addressing this point (as is also acknowledged by reviewer 3), and thus complying with best practice in the field, the reviewer now requests an update of all other tools, claiming that this would be needed for a meaningful comparison.

*We appreciate critical (but fair) and constructive comments that help improving the quality of work. However, in this case, we are convinced that this request is unreasonable and unjustified. It is beyond the scope of our work (and responsibility) to update the tools of other developers, which is even technically impossible to achieve as we had already stated in our first response. To reiterate, many tools tested in CAMI do not offer this option. Most importantly, for a fair comparison, it is critical that comparisons are made using reference sequences that are NOT included in the reference databases of taxonomic profiling tools, a design that was specifically followed in the CAMI challenge. Importantly, the reference genome database used in mOTUs2 was generated **before** the CAMI challenge closed and thus does not constitute an unfair advantage.*

Also, regarding the criticism that good practice would require to add functionality for rebuilding/updating databases, we did implement a tutorial for adding new genomes into the mOTU database as we had clearly mentioned as an action taken in response to the first round of revisions. Unfortunately, this was not acknowledged or commented on by the reviewer at all.

In addition to these general remarks, we provide more detailed responses in the following.

This affects the following issues:

(1) I truly appreciate the efforts by the authors to compare on CAMI data. As far as I understand from the description in the manuscript, the authors ran their analysis with recent data, but did not rerun the other analyses with most

recent data – thus any difference in performance could be due to two reasons: (i) database size or (ii) conceptual advances of mOTUv2. Given the current setup these effects cannot be disentangled and no conclusion can be drawn.

Response:

By requesting to update all other tools and re-run the comparisons, the reviewer is now diverting from his/her original view of CAMI serving as the gold standard / established community standard for a fair comparison of metagenomic tools. Importantly, it is essential that the benchmark dataset contains sequences that are not available to any of the evaluated tools. It is even technically impossible to update the databases of all other tools, which would require more effort than redoing CAMI completely, which we also mentioned in our previous response. Tools are never developed at the same time nor are their resources standardized (in any field of bioinformatics). Thus, it is a given that both database and algorithms were evaluated simultaneously.

Still, strong conclusions can be drawn on the empirical performance of tools as they are deployed, without extensive updates or tweaks that can only be done by highly specialised experts. Most probably, such conclusions are of direct interest to the vast majority of readers and researchers. Disentangling algorithmic superiority - which we have never claimed for mOTUs2 - from empirically superior performance on a standardised benchmark data set is arguably of much lower interest.

The reviewer is also mistaken by stating that the database underlying mOTUs2 provides an unfair advantage due to being more recent than those used by other tools. In fact, the download of genomes used in mOTUs2 precedes the closing date of the CAMI competition. In CAMI, tools are typically used as available and as applied by the majority of users.

Taken together, we are confident that by evaluating the empirical performance of mOTUs2 in extensive benchmarks (Fig. 2) including the use of the CAMI dataset, we comply with state-of-the-art scientific practice and community standards for comparing metagenomic tools.

(2) I appreciate the plans by the authors to update mOTU v2 regularly. This commitment should be explicitly stated in the manuscript: 'The authors plan to provide X updates for mOTU v2 per year'. The authors and I appear to agree that without these updates, mOTU v2 would soon be outdated.

Response:

Note that for expert users, we already implemented a tutorial on how to update the current database as had been clearly stated as an action taken in response to the previous request. While we do not agree that specific update frequencies should be stated, we did include a sentence in the manuscript implying our intention to regularly update the tool.

Action taken:

We added a sentence (line: 370-371) that reads: "Regular updates of the of the mOTU database will be made available at: <http://motu-tool.org>."

(3) I understand the difficulties of the authors to build a kraken database. While the limitations described by the authors do appear solvable to me (I happen to know that Zurich for instance has an excellent support in scientific computations that should not have trouble providing such a machine which could also be easily be rented in the cloud). Still, there are several publications (centrifuge, krakenhll, ganon) that show improved variants of the kraken approach that more easily cope with larger databases. If infrastructural problems do persist that may be a sound alternative to show that the reason for better performance is not solely based on larger databases.

Response:

It appears, the reviewer keeps misunderstanding our reasoning. To reiterate, the preprint article (now published) referenced by the reviewer (<https://doi.org/10.1101/304972>) demonstrates that updating databases for kraken or similar tools is not a trivial task. In fact, considering that the subject of this publication is on updating databases and comparing the results shows that this amount of work is a research project on its own. Furthermore, the requested

updates would not only require an intense infrastructural effort, but development efforts on tools other than our own one. More importantly, though, the minikraken database used in our work is already based on more recent genomes than those used in mOTUs2, which uses merely 600 Mb of sequences as a reference database demonstrating that size alone is not a meaningful parameter for performance. We also clearly describe several (algorithmic) improvements that have contributed to the overall performance of mOTUs that go beyond the extended database in version 2.

(4) Again, I appreciate the efforts by the authors, but my original questions whether integration with other tool results improves on mOTU v2 remains unanswered. Further, the comparison would be most useful if carried out on most recent database versions.

Response:

Here again, the reviewer is asking for development efforts on other tools than the ones we are aiming to publish, which in our view is an unreasonable request. In particular, it has already been shown in the original publication of MetaMeta (a tool that integrates several taxonomic profilers and the one referenced by this reviewer) that even the first version of mOTUs as a standalone tool performed better than MetaMeta (even though mOTUs v1 was integrated into MetaMeta). In response to this request, we pointed that out and showed that also mOTUs2 outperforms MetaMeta (and its previous version mOTUs1). As discussed in our previous answer to the reviewer, we are convinced that this request will not yield any meaningful result and consider it unjustified for the reasons mentioned above, in particular as it is reiterated without reacting to the reasoning detailed in our previous response to the same request.

REVIEWERS' COMMENTS:

Reviewer #2 (Remarks to the Author):

Most metagenomic profiling tools are composed by a piece of software implementing a computational method that relies on a custom reference database provided by the authors of the tool. The former cannot be used without the latter, and reconstructing or updating the database is usually not a trivial task that requires a considerable effort, so that users in practice tend to rely on the original reference database. It is also a common practice, that should be discouraged in any possible way, that authors, once they have a publication in a high impact journal, very rarely update the database, a fact that quickly render their tool useless for any practical purpose. These zombie tools remain in the public domain, and constitute a problem both for users and developers.

The issue between Reviewer #1 and the authors is very clearly defined by Reviewer #1 at the beginning of his report, and reduces to the following question: when comparing two computational methods, should we try to disentangle the performance of the computational algorithm from the completeness of the reference database, or assess the whole package, i.e software plus reference database? I tend to share the view of the authors that the second alternative is the correct one. As a user of metagenomic profiling tools, I am not interested in this distinction. Given that, even if a procedure is provided by the authors, building a new reference database for a given tool usually requires more effort than analyzing the data themselves, I would choose the tool that guarantees the best out-of-the-box performance, and I am sure that most researchers would do the same. As an author of software tools, I feel that it is not my responsibility to maintain other's people work, and I feel very frustrated when I have to demonstrate that my software works better than some other tool if the authors had updated the reference database.

However, I agree with Reviewer #1 in one point, namely point (2). In order to guarantee that their tool does not follow the same fate of the others, and become obsolete as soon as the manuscript is accepted, they should commit to update the reference database regularly, even if I am not sure how in practice we could check that they actually do it.

Response to reviewer #2 evaluating comments by reviewer #1

Reviewer #2 (Remarks to the Author):

Most metagenomic profiling tools are composed by a piece of software implementing a computational method that relies on a custom reference database provided by the authors of the tool. The former cannot be used without the latter, and reconstructing or updating the database is usually not a trivial task that requires a considerable effort, so that users in practice tend to rely on the original reference database. It is also a common practice, that should be discouraged in any possible way, that authors, once they have a publication in a high impact journal, very rarely update the database, a fact that quickly render their tool useless for any practical purpose. These zombie tools remain in the public domain, and constitute a problem both for users and developers.

The issue between Reviewer #1 and the authors is very clearly defined by Reviewer #1 at the beginning of his report, and reduces to the following question: when comparing two computational methods, should we try to disentangle the performance of the computational algorithm from the completeness of the reference database, or assess the whole package, i.e software plus reference database? I tend to share the view of the authors that the second alternative is the correct one. As a user of metagenomic profiling tools, I am not interested in this distinction. Given that, even if a procedure is provided by the authors, building a new reference database for a given tool usually requires more effort than analyzing the data themselves, I would choose the tool that guarantees the best out-of-the-box performance, and I am sure that most researchers would do the same. As an author of software tools, I feel that it is not my responsibility to maintain other's people work, and I feel very frustrated when I have to demonstrate that my software works better than some other tool if the authors had updated the reference database.

However, I agree with Reviewer #1 in one point, namely point (2). In order to guarantee that their tool does not follow the same fate of the others, and become obsolete as soon as the manuscript is accepted, they should commit to update the reference database regularly, even if I am not sure how in practice we could check that they actually do it.

Response:

We are grateful for reviewer #2's evaluation of our responses to reviewer #1. We are particularly pleased to see that (s)he shares our view that software tools should be assessed for their performance as a whole package and that it is not in the responsibility of a tool developer to maintain other people's work.

As previously stated in our response to the additional comments by reviewer #1 below, we added a sentence to the manuscript that reads "Regular updates of the of the mOTU database will be made available at: <http://motu-tool.org>." to indicate our commitment in maintaining the tool.